# Particle-Guided Diffusion Models for Partial Differential Equations

Andrew Millard [1]   Fredrik Lindsten [1]   Zheng Zhao [1]

## Abstract

We introduce a guided stochastic sampling method that augments sampling from diffusion models with physics-based guidance derived from partial differential equation (PDE) residuals and observational constraints, ensuring generated samples remain physically admissible. We embed this sampling procedure within a new Sequential Monte Carlo (SMC) framework, yielding a scalable generative PDE solver. Across multiple benchmark PDE systems as well as multiphysics and interacting PDE systems, our method produces solution fields with lower numerical error than existing state-of-the-art generative methods.

## 1. Introduction and Related Work

Partial differential equations (PDEs, Evans, 2022; Strauss, 2007) form the foundation of scientific and engineering models (Ames, 2016), governing phenomena such as fluid flow (Naz et al., 2008), heat transport (Crank & Nicolson, 1947), elasticity (Smith, 1990), electromagnetics (Maxwell, 1865), and the ability to accurately solve PDEs is thus a key enabler for scientific and engineering progress across a broad range of applications (Zachmanoglou & Thoe, 1986). Classical numerical solvers, such as including finite difference (LeVeque, 2007), finite element (Johnson, 2009), and spectral methods (Trefethen, 2000), provide reliable and well-understood approximations, but their computational cost grows rapidly with problem size, nonlinear dynamics, and parameter variability. As a result, simulations in high resolution or across large parameter spaces remain prohibitively expensive for real-time, uncertainty-aware, or many-query settings.

Recent efforts have explored machine learning techniques to accelerate PDE solution. Physics-informed neural networks (PINNs, Cai et al., 2021a;b; Raissi et al., 2019) embed the governing equations into a neural-network loss function and can learn solution fields in addition with observation data. Operator-learning models, such as Fourier Neural Operators (Li et al., 2021; 2023) and DeepONets (Lu et al., 2019; 2021), aim to learn mappings from forcing terms or parameters to full-solution fields and enable fast and flexible inference once trained. Neural surrogates, e.g., (Kochkov et al., 2021; Sanchez-Gonzalez et al., 2020), replace expensive numerical methods with models learned from simulations and reduced-order models (Tripathy & Bilionis, 2018; White et al., 2019; Eason & Cremaschi, 2014) further accelerate computation by approximating fine-resolution solutions using coarse representations refined by learned corrections. However, these ML-based PDE solvers produce deterministic predictions (Huang et al., 2024) and may require large datasets or costly hyperparameter tuning. Ensuring physical consistency and robustness across varying boundary conditions and parameter regimes remains challenging, motivating the exploration of probabilistic and generative approaches for forward PDE modeling.

In parallel, flow-based generative models based on diffusion (Song et al., 2021) and stochastic interpolants (Albergo et al., 2025) have emerged as state-of-the-art tools for sampling from complex, high-dimensional distributions. These models gradually transform noise into structured samples via stochastic differential equations, enabling diverse and uncertainty-aware generation in domains such as images (Rombach et al., 2022; Dhariwal & Nichol, 2021), audio (Kong et al., 2021; Huang et al., 2023; Liu et al., 2023), and molecular design (Guo et al., 2024; Yim et al., 2024). Their probabilistic formulation allows sampling from full posterior distributions rather than returning a single point estimate. These properties make diffusion models a promising candidate for scientific simulation, where multiple physically valid solutions may exist.

### 1.1. Related Work

Within the generative modeling literature, various methods have been applied to solving PDEs which can be broadly separated into two different categories: *data driven* and *physics-informed*.

The data driven approach models the underlying relationship between inputs and outputs by discovering correla-

---

[1]Department of Computer and Information Science, Linköping University, Linköping, Sweden. Correspondence to: Zheng Zhao <zheng.zhao@liu.se>.

*Proceedings of the 43rd International Conference on Machine Learning*, Seoul, South Korea. PMLR 306, 2026. Copyright 2026 by the author(s).

tions and patterns directly from the data without explicitly prescribing physical laws. In the context of PDEs, many generative methods have attempted to use the data driven approach. For instance, variational Autoencoders and generative adversarial networks have been used extensively to model PDEs (Gonzalez & Balajewicz, 2018; Creswell et al., 2018; Barati Farimani et al., 2017; Mirza & Osindero, 2014). Flow-based generative models have also found success recently in this area, with applications in fluid field prediction (Yang & Sommer, 2023; Kohl et al., 2024; Liu & Thuerey, 2024), weather forecasting (Price et al., 2025), 3-dimensional turbulent flows (Molinaro et al., 2024), modeling buoyancy (Li et al., 2025).

Physics-informed approaches are common in the PINN literature (Cai et al., 2021a;b; Raissi et al., 2019) but have seen less common adoption by the generative modeling community.

Lienen et al. (2024) utilize Dirichlet boundary conditions as guidance when sampling, but do not incorporate the full equations as part of a loss function, therefore it is still primarily a data driven approach. Shysheya et al. (2024); Rozet & Louppe (2023) use diffusion models trained on PDE data in combination with inference-time ("zero shot") guidance for data assimilation, i.e., to condition the generation on sparse measurements. Closest to our work is DiffusionPDE (Huang et al., 2024) which similarly uses guidance to incorporate the governing equations into the likelihood during the sampling process.

The guidance methods uses in these framework are not specific to PDEs, and many methods have been developed for controlled generation, predominantly in computer vision (see, Daras et al., 2024; Chung et al., 2025, for recent surveys). Motivated by the need for diversity and statistical consistency, several recent contribution incorporate Sequential Monte Carlo (SMC, Wu et al., 2023; Cardoso et al., 2024; Kelvinius et al., 2025; Zhao, 2026) and Markov Chain Monte Carlo (Dang et al., 2026; Janati et al., 2025; Corenflos et al., 2025; Kalaivanan et al., 2025) in the guidance procedure. However, we have yet to see an adaptation of such methodology to physics-informed regularization of generative PDE solvers.

### 1.2. Contributions

We propose a new method for guidance of pretrained generative PDE solvers leveraging SMC for improved efficiency. This enables data assimilation as well as physics-informed regularization for solving constrained PDE systems.

However, while existing SMC-based guidance methods in computer vision and other domains (Wu et al., 2023; Cardoso et al., 2024; Stevens et al., 2025; Dou & Song, 2024; Kelvinius et al., 2025; Zhao et al., 2025) are primarily moti-

vated by statistical consistency, we challenge this interpretation. While these methods indeed show improved empirical performance compared to their non-SMC counterparts, we argue that this is *despite the fact that the SMC algorithms are highly degenerate when applied to conditional generative models*. In other words, we argue that existing SMC-based guidance methods for high-dimensional generative models are effective, not because they achieve a high effective sample size, but due to their inherent "multiple try" sampling nature. We elaborate on this interpretation in Section 3.5.

Based on this insight, our contributions are:

- We develop an SMC-based guidance method for data assimilation and physics-informed regularization for solving constrained PDE systems.

- We propose to use a second order stochastic proposal augmented with PDE guidance (referred to as the SOSaG proposal) for better integration of the diffusion models.

- While the basic SMC-formulation is theoretically consistent, we show that it's incompatible with the efficient SOSaG proposal. We therefore propose a generalization of the SMC framework that trades statistical consistency for improved empirical performance, effectively resulting in a hybrid "vanilla guidance" and SMC approach.

- We demonstrate that this new method outperforms other guided sampling approaches (Huang et al., 2024) on multiple benchmark datasets, as well as two and three species interacting PDE systems.

## 2. Problem Formulation and Background

### 2.1. Partial Differential Equations

We consider the case for both static and time dependent PDEs. In the static case, for a spatial domain $\Omega$ a PDE can be written as

$$
\begin{aligned}
f(c, a, u) &= 0, \quad c \in \Omega, \\
u(c) &= g(c), \quad c \in \partial\Omega,
\end{aligned} \tag{1}
$$

where $c$ is a spatial coordinate, $u \in \mathcal{U}$ is the solution field of the PDE, and $a \in \mathcal{A}$ is a coefficient field that describes some properties of the system which influence the PDE. For example, in Darcy flow, one solves for the pressure field $u(c)$ given the permeability field $a(c)$ of a specific medium $-\nabla \cdot (a(c)\nabla u(c)) = s(c)$ and $s(c)$ is a source term such as a forcing function. For a time-dependent PDE defined over a time horizon $[0, \mathcal{T}]$, we write

$$
\begin{aligned}
f(c, \tau, a, u) &= 0, \quad c \in \Omega, \quad \tau \in [0, \mathcal{T}], \\
u(c, \tau) &= g(c, \tau), \quad c \in \partial\Omega, \quad \tau \in [0, \mathcal{T}], \\
u(c, 0) &= a(c),
\end{aligned} \tag{2}
$$

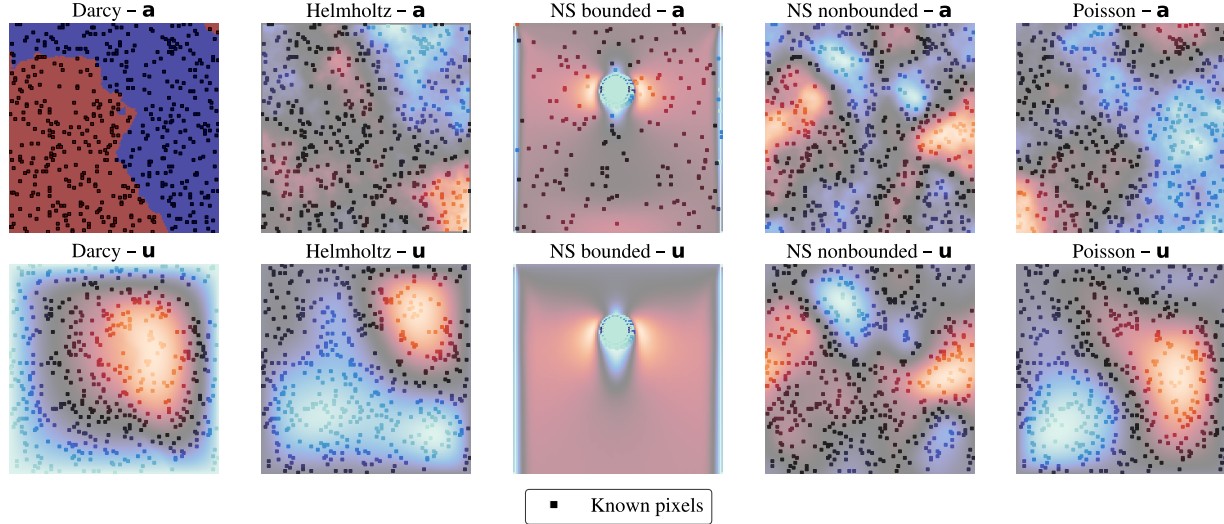

Darcy – **a** · Helmholtz – **a** · NS bounded – **a** · NS nonbounded – **a** · Poisson – **a**

Darcy – **u** · Helmholtz – **u** · NS bounded – **u** · NS nonbounded – **u** · Poisson – **u**

■ Known pixels

*Figure 1.* Contour plot of the ground truth parameters overlaid with the observed values (highlighted pixels) for each PDE. The top row shows the coefficient field and the bottom row shows the solution field.

where $\tau$ denotes PDE time (to be distinguished from the denoising diffusion time $t$). Here, $u$ is the solution field and $a$ specifies the initial condition and/or PDE coefficients. As an example of such a dynamic system, we consider the incompressible Navier–Stokes (INS) equations, where the solution field is the velocity $u(c, \tau) = v(c, \tau)$, together with a pressure field $p(c, \tau)$, viscosity $\nu$, and external forcing $q(c, \tau)$. In this setting, the initial condition is given by $a(c) = v(c, 0)$, and our goal is to recover $a$ and/or the solution at a later time $u_\mathcal{T} = u(\cdot, \mathcal{T})$ from sparse observations.

### 2.2. Sampling for Generative Diffusion Models

Generative diffusion models (Sohl-Dickstein et al., 2015; Song et al., 2021; Ho et al., 2020; Cao et al., 2024; Karras et al., 2022) provide a probabilistic framework for sampling from complex, high-dimensional data distributions $p_{\text{data}}(x)$.[1] Over many time steps, $p_{\text{data}}(x)$ is gradually corrupted by adding Gaussian noise with standard deviation $\sigma$, where $\sigma$ is obtained from a predetermined noise schedule such that $\sigma_t \in [0, \sigma_{\max}]$ and $t \in [0, T]$. Therefore, $p(x, \sigma_{\max}) = p_{\text{ref}}(x) \approx \mathcal{N}(0, \sigma_{\max}^2 I)$. In order to generate samples that approximate $p_{\text{data}}(x)$, we draw a sample $x_T \sim \mathcal{N}(0, \sigma_{\max}^2 I)$ and gradually denoise the samples with $\sigma_t$ so that $x_t \sim p_t(x_t, \sigma_t)$ and $x_0 \sim p_{\text{data}}(x)$.

The probability flow ordinary differential equation (ODE) describes how a sample can be denoised (Song et al., 2021; Karras et al., 2022) in reverse-time according to

$$dx_t = -\dot{\sigma}_t \sigma_t \nabla_{x_t} \log p_t(x_t, \sigma_t) dt, \quad t \in [0, T], \quad (3)$$

---

[1]For consistency, we follow the EDM formulation (Karras et al., 2022), but our methodology can be generalized to other flow-based generative models (Albergo et al., 2025; Lipman et al., 2023).

where $\nabla_x \log p_t(x_t, \sigma_t)$ is the score function, and the forward noising process is considered a Brownian motion as in Karras et al. (2022). It has previously been proposed (Karras et al., 2022) to learn the denoising function $D_\theta(x; \sigma)$ such that the score function can be estimated by:

$$\nabla_x \log p_t(x_t, \sigma_t) = \frac{x_t - D_\theta(x_t, \sigma_t)}{\sigma_t^2}. \quad (4)$$

Here, $D_\theta$ is a neural network trained to predict the denoised version of a noisy sample at a given noise level. Consequently, the score is proportional to the difference between the noisy sample and its denoised estimate, yielding an implicit estimate of the added noise.

An alternative to ODE sampling is to simulate the reverse-time stochastic differential equation (SDE), described by

$$dx_t = -2\dot{\sigma}_t \sigma_t \nabla_x \log p_t(x_t, \sigma_t) dt + \sqrt{2\dot{\sigma}_t \sigma_t} \, dW_t, \quad (5)$$

where $W_t$ is a Brownian motion. The derivation of the ODE and SDE equivalence follows from matching the path distribution. Again, a denoising neural network $D_\theta$ is trained which in turn is used to approximate the score $\nabla_x \log p_t(x, \sigma_t)$, enabling approximate inversion of the original noising process through a reverse-time SDE.

### 2.3. Stochastic Guided Sampling for Solving PDEs

In this work we propose to solve PDE equations by using a diffusion model, pretrained on the joint distribution of coefficients $a$ and solutions $u$, similar to Huang et al. (2024). This model acts as a prior distribution and we then use PDE residuals as guidance during inference. Thus, the samples generated by the diffusion model are the tuples $x = (a, u)$

where

$$x \in \mathcal{X}, \qquad \mathcal{X} = \mathcal{A} \times \mathcal{U}, \tag{6}$$

i.e., concatenating the PDE coefficients and solution, and we denote the prior distribution over these tuples by $p_{\text{data}}$.

However, naively solving a PDE using this approach is fundamentally no different compared to just generating an image (see, e.g., Yang & Sommer, 2023), ignoring two crucial information: the governing PDE equation, and often times, sparse observations $a_{obs}, u_{obs}$ of the coefficients and solution. We thus propose to incorporate these information as a condition $y$ and aim to sample from a conditional/posterior diffusion model. The sampling can be achieved by simulating the conditional diffusion model

$$dx_t = \underbrace{-2\dot{\sigma}_t\sigma_t\nabla_{x_t}\log p_t^\theta(x_t, \sigma_t)dt}_{\text{Score term}}$$
$$\underbrace{-2\dot{\sigma}_t\sigma_t\nabla_{x_t}\log p_t^\theta(y \mid x_t, \sigma_t))dt}_{\text{Guidance term}} + \sqrt{2\dot{\sigma}_t\sigma_t}\,dW_t. \tag{7}$$

In addition to the score term, we note that (7) now has a gradient term based on the conditional information which we call the *guidance* term. This extra term will tilt the unconditional diffusion model at time $t = 0$, to the posterior distribution

$$p_\theta(x \mid y) \propto p(y \mid x)p_\theta(x), \tag{8}$$

where $p_\theta(x) \approx p_{\text{data}}$ is given by a pre-trained diffusion model prior. However, exactly targeting this posterior would require computing the intermediate likelihood $p_t^\theta(y \mid x_t, \sigma_t) = \int p(y \mid x_0)p_\theta(x_0 \mid x_t)dx_t$ which is in general intractable. Different guidance methods (Dou & Song, 2024; Zhao et al., 2025; Daras et al., 2024; Chung et al., 2025) *approximate* this intermediate likelihood in different ways. In the next section we introduce Sequential Monte carlo (SMC) which leverages interacting particles as a way to correct the approximation and to form the posterior distribution.

### 2.4. Sequential Monte Carlo

SMC methods (e.g., Chopin et al., 2020; Naesseth et al., 2019) provide a principled particle-based framework for sampling from a sequence of distributions, and naturally align with the sequential denoising procedure of generative diffusion models. Previous works have used the SMC framework as a conditional sampling algorithm for diffusion models (Wu et al., 2023; Cardoso et al., 2024; Stevens et al., 2025; Dou & Song, 2024; Kelvinius et al., 2025; Zhao et al., 2025). In the previous sections $t$ was used when discussing continuous time processes. In reality, when simulating these processes we approximate them in discrete time $k$ and therefore we shall be using $k$ in the following sections.

SMC evolves a population of $N$ particles $\{x_k^{(i)}\}_{i=1}^N$ across $k = 0, 2, \ldots, K$ time steps. The key components of SMC

are the proposal distributions/Markov kernels $M_K(x_K)$, $\{M_{k-1}(x_{k-1} \mid x_k)\}_{k=1}^K$ and the weighting/potential functions $G_K(x_K)$, $\{G_{k-1}(x_k, x_{k-1})\}_{k=1}^K$. Initially, we draw $N$ samples from our initial proposal $x_K^{(i)} \sim M_K = \mathcal{N}(0, I)$ and then weight them according to an initial potential function $w_K^{(i)} \propto G_K(x_K^{(i)})$. For $K$ iterations, we then sequentially propose $x_{k-1}^{(i)} \sim M_{k-1}(x_{k-1}^{(i)} \mid x_k^{(i)})$, weight samples $w_{k-1}^{(i)} \propto w_k^{(i)} G_{k-1}(x_k^{(i)}, x_{k-1}^{(i)})$, and resample (Douc & Cappé, 2005) if our effective sample size (ESS) drops below a certain threshold $N_{eff}$. An overview of this process is given in Appendix A.1. At each iteration $k$ the SMC algorithm generates a weighted particle population $\{(x_k^{(i)}, w_k^{(i)}\}_{i=1}^N$ that provides an empirical approximation of the target distribution

$$\nu_k(x_{k:K}) \propto G_K(x_K)M_K(x_K)$$
$$\times \prod_{j=k+1}^K G_{j-1}(x_j, x_{j-1})M_{j-1}(x_{j-1} \mid x_j). \tag{9}$$

We apply the SMC framework to sample the posterior distribution in Equation (8) by choosing the proposals and weight functions $\{M_k, G_k\}_{k=0}^K$ in such a way that the final *marginal* $\nu_0(x_0) = p_\theta(x_0 \mid y)$. As long as this requirement is fulfilled, the algorithm will provide a consistent (as $N \to \infty$) Monte Carlo estimate of the target at the final time point regardless of the intermediate marginals. In practice, however, these design choices influence the statistical efficiency of the algorithm (e.g., the weight variance).

## 3. Methodology

### 3.1. PDE Residual Likelihoods

In the context of PDEs, we can express our likelihood as the mean squared error (MSE) of the sparse observations for both $a$ and $u$ as well as the PDE equation residuals. That is, we define the *PDE residual likelihood*:

$$\log p(y \mid x) = -\beta\left(\frac{1}{n}||u_{obs} - \mathcal{M}_u \odot x||_2^2\right)$$
$$-\gamma\left(\frac{1}{n}||a_{obs} - \mathcal{M}_a \odot x||_2^2\right)$$
$$-\omega\left(\frac{1}{m}||f(c, \tau, x)||_2^2\right) + c, \tag{10}$$

where $\mathcal{M}_{u,a}$ are the corresponding binary masks indicating the spatial coordinates of the observations, $n$ is the number of observations, $m$ is the number of pixels and $\odot$ is the Hadamard/element-wise product. This likelihood jointly incorporates the governing PDE as a physics constrain, and the sparse observation as a data constrain.

To enable approximate likelihood evaluations at intermediate time steps we follow Wu et al. (2023) and evaluate the

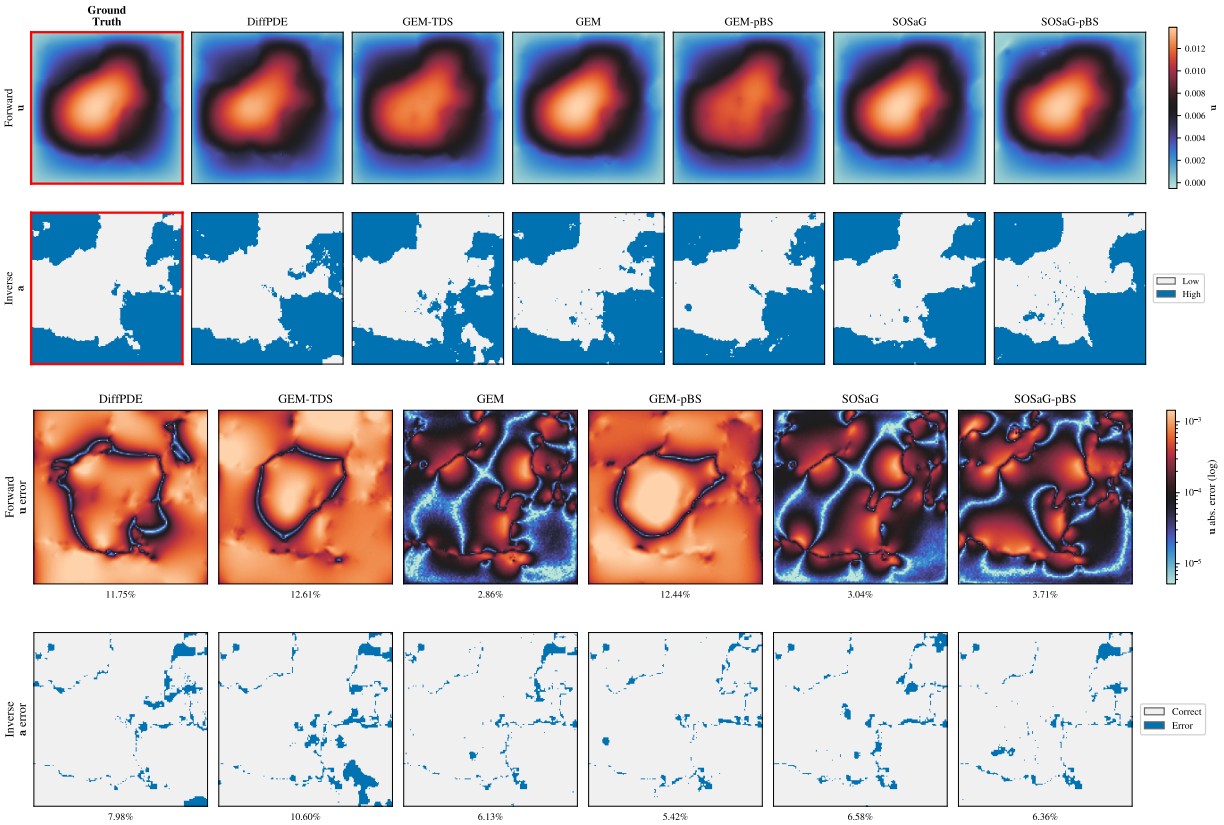

*Figure 2.* The top two rows show contour plot of the generated PDE solutions and coefficients using different methods. The bottom two rows show the corresponding (relative) error when compared to the ground truth (GT). We especially observe that the most erroneous part are around the edges. This is consistent with the common problem of diffusion models that they tend to smooth out the high-frequency information of the generated samples.

likelihood at reconstructed samples. That is, we define the intermediate approximate likelihood at diffusion time $t$ as:

$$p_t^\theta(y \mid x_t, \sigma_t) = \int p(y \mid x_0) p_\theta(x_0 \mid x_t) dx_0$$

$$\approx p(y \mid x_0 = D_\theta(x, \sigma_t)) =: \tilde{p}_\theta(y \mid x_t), \quad (11)$$

and as a result, at time 0 we have $\tilde{p}_\theta(y \mid x_0) = p(y \mid x_0)$.

In the following sections, we show how we can incorporate this likelihood along with different proposals into the SMC framework, by designing suitable $\{M_k, G_k\}_{k=0}^K$.

### 3.2. Guided Euler–Maruyama as a Proposal for an SMC Sampler

To numerically sample the SDE (5), a common approximation is the Euler–Maruyama method:

$$x_{k-1} = x_k - (\sigma_{k-1}^2 - \sigma_k^2) \nabla_{x_k} \log p_{k-1}(x_k, \sigma_k)$$

$$+ \sqrt{\sigma_{k-1}^2 - \sigma_k^2} z, \quad (12)$$

where $z \sim \mathcal{N}(0, I)$, and the discretization time interval is implicitly accounted in the schedule sequence $\sigma_k$.

We denote this transition as $p_\theta(x_{k-1} \mid x_k) = \mathcal{N}(x_k \mid \mu_{\text{model}}, (\sigma_{k-1}^2 - \sigma_k^2) I)$. Using the Euler–Maruyama method to simulate from $k = K$ to $k = 0$ we obtain a trajectory distributed according to $p_\theta(x_{0:K})$.

If we use the guided score (7) in place of the unconditional score in equation (12), we obtain the guided Euler-Maruyama (GEM) update:

$$x_{k-1} = x_k - (\sigma_{k-1}^2 - \sigma_k^2) \frac{x_k - D_\theta(x_k, \sigma_k)}{\sigma_k^2}$$

$$- (\sigma_{k-1}^2 - \sigma_k^2) \nabla_{x_k} \log \tilde{p}_\theta(y \mid x_k)$$

$$+ \sqrt{\sigma_{k-1}^2 - \sigma_k^2} z. \quad (13)$$

Pseudocode for this method is shown in Appendix A.4.1. The GEM update (13) defines a Markov transition kernel $\tilde{p}_\theta(x_{k-1} \mid x_k, y) = \mathcal{N}(x_k \mid \mu_{\text{prop}}, (\sigma_{k-1}^2 - \sigma_k^2) I)$ that can be used as a proposal for SMC. Specifically, we choose the proposal $M_{k-1}(x_{k-1} \mid x_k) = \tilde{p}_\theta(x_{k-1} \mid x_k, y)$, and the weight update

$$G_{k-1}(x_k, x_{k-1}) = \frac{\tilde{p}_\theta(y \mid x_{k-1})}{\tilde{p}_\theta(y \mid x_k)} \frac{p_\theta(x_{k-1} \mid x_k)}{\tilde{p}_\theta(x_{k-1} \mid x_k, y)} \quad (14)$$

This was used in the Twisted Diffusion Sampler (TDS, Wu et al., 2023), and we can verify using the FK formula (9) and substituting in our proposal and weight update that, the marginal distribution we target at each iteration is:

$$\nu_{k-1}(x_{k-1:K}) \propto \tilde{p}_\theta(y \mid x_{k-1}) p_\theta(x_{k-1:K})$$
$$\nu_{0:K}(x_{0:K}) = p_\theta(x_{0:K} \mid y). \tag{15}$$

### 3.3. Second-Order Stochastic Guided Proposal

Although Euler–Maruyama is common, it is not the most efficient SDE integrator and Karras et al. (2022) propose using 2nd order stochastic sampling process. First, the noise scale is jittered and then the ODE dynamics via the denoiser are run, i.e. we run the following set of equations:

$$\hat{\sigma}_k \leftarrow \sigma_k + \gamma_k\,\sigma_k, \tag{16}$$

$$\hat{x}_k = x_k + \sqrt{\hat{\sigma}_k^2 - \sigma_k^2}\,\psi, \quad \psi \sim \mathcal{N}(0, I), \tag{17}$$

$$x_{k-1} = \hat{x}_k + \left(\sigma_{k-1}^2 - \hat{\sigma}_k^2\right)\frac{\hat{x}_k - D_\theta(\hat{x}_k, \hat{\sigma}_k)}{\hat{\sigma}_k^2}, \tag{18}$$

where $\gamma_k$ is a constant used to reach a higher noise level. The full details of this can be found in Karras et al. (2022). Notice how this algorithm adds noise before computing the denoised mean. Including guidance information as outlined previously, (18) becomes:

$$x_{k-1} = \hat{x}_k + \left(\sigma_{k-1}^2 - \hat{\sigma}_k^2\right)\frac{\hat{x}_k - D_\theta(\hat{x}_k, \hat{\sigma}_k)}{\hat{\sigma}_k^2}$$
$$- \left(\sigma_{k-1}^2 - \hat{\sigma}_k^2\right)\nabla_{\hat{x}_k} \log \tilde{p}_\theta(y \mid \hat{x}_k). \tag{19}$$

This method uses a second-order correction method to improve sampling quality. This was also implemented in the DiffusionPDE framework (Huang et al., 2024), albeit with a deterministic (ODE-based) formulation. We also use the 2nd order correction as part of our proposal but, importantly, with the stochastic jittering. We refer to this proposal as the Second-Order Stochastic Guided (SOSaG) proposal.

Although the SOSaG integrator is shown to be empirically powerful, it is not straightforward to incorporate it as the proposal $M$ in SMC. The reason is that the weight computation in equation (14) requires a point-evaluable proposal density which is not the case for SOSaG. Clearly, unlike Euler–Maruyama, now $x_{k-1}$ conditioned on $x_k$ does not retain the Gaussianity after passing through the non-linear denoiser. However, we can instead work on a new Feynman–Kac model $\overline{\nu}_{0:K}$ with components defined in a "bootstrap" fashion, which we call pseudo-bootstrap (pBS):

$$\overline{M}_{k-1}(x_{k-1} \mid x_k) = \overline{p}_\theta(x_{k-1} \mid x_k, y),$$
$$\overline{G}_{k-1}(x_k, x_{k-1}) = \frac{\tilde{p}_\theta(y \mid x_{k-1})}{\tilde{p}_\theta(y \mid x_k)},$$

where $\overline{p}_\theta(x_{k-1} \mid x_k, y)$ denotes the distribution of SOSaG proposal described in equation (19). At the terminal time $k = 0$, this new model recovers $\overline{\nu}_0(x_0) = \overline{p}_\theta(x_0 \mid y)\,p_\theta(y \mid x_0)$, where $\overline{p}_\theta(x_0 \mid y)$ is an approximate posterior distribution obtained by simulating the SOSaG proposal without SMC correction (see Appendix A.3 for the derivation). Although this new target is no longer the same as the original in (15), it is motivated in this PDE context, as we explain in the following.

### 3.4. Bridging between pBS and PINNs

When using the pBS framework, our target distribution is altered so it is multiplied by an extra likelihood factor. Theoretically, we could choose to use any multiple of the likelihood depending on our belief in the diffusion prior and the PDE residual likelihood:

$$\overline{\nu}_{0:K}(x_{0:K}) \propto \overline{p}_\theta(x_0 \mid y)\tilde{p}_\theta(y \mid x_0)^\rho, \tag{20}$$

where $\overline{p}_\theta(x_0 \mid y)$ is the final marginal of the SOSaG proposal, $\rho$ is often called the *tempering* parameter (Neal, 2001; Buchholz et al., 2021). In our case, the temperature is simply $\rho = 1$. As $\rho$ increases, the target is dominated by the likelihood which approaches a pure maximum likelihood estimation (MLE) task which is the framework PINNs use. This is essentially changing the precision of the PDE residual likelihood. This also gives us a way to control our prior belief. It is unlikely that our diffusion prior will perfectly model probability distribution of the PDE governing the data. Therefore, a traditional SMC sampling method will approach the desired target, but this target may not be the optimal-in-practice solution to our problem. Empirically, we find that this produces better results and therefore it motivates that the "true target" is different than the one converged upon by Equation (15).

### 3.5. SMC and Evolutionary Algorithms

A common criticism of SMC, especially when sampling from diffusion models is the effective sample size (ESS) degenerates as $k \to 0$ (Wu et al., 2023; Corenflos et al., 2025; Zhao, 2026), often only leaving a single particle with a weight that contributes towards the target estimate, despite having relatively high ESS at the beginning. However conversely, it has also largely been empirically shown that SMC samplers still work well albeit the particle coalescence issue. To generate diverse samples, Dou & Song (2024); Trippe et al. (2023) run the SMC independently many times, usually with a low number of particles and keeping a single particle per iteration. This results in a biased, but empirically effective, approximation of the SMC target distribution.

The efficacy of this approach can be understood as a form of evolutionary algorithm (Bäck & Schwefel, 1993; Yu & Gen, 2010), where the weights represent the survival probabilities,

*Table 1.* Comparison of different models on five PDE problems (in $L_2$ relative error, except in the Darcy inverse problem where error rate is reported).

| Method | Darcy | | Poisson | | Helmholtz | | NS | | NS (BCs) | |
|---|---|---|---|---|---|---|---|---|---|---|
| | **Fwd** | **Inv** | **Fwd** | **Inv** | **Fwd** | **Inv** | **Fwd** | **Inv** | **Fwd** | **Inv** |
| *ODE-Based Methods* | | | | | | | | | | |
| DiffPDE | 5.58 | 8.31 | 8.67 | **19.88** | 17.49 | 19.14 | 4.27 | **9.34** | 2.78 | 3.43 |
| *1st Order Stochastic Methods* | | | | | | | | | | |
| GEM | 5.28 | 7.34 | **6.53** | 22.28 | 11.22 | 19.30 | **4.16** | 10.10 | 2.31 | 2.36 |
| GEM-TDS | 5.52 | 9.00 | 16.44 | 35.27 | 21.75 | 22.69 | 4.18 | 10.85 | 2.96 | 3.52 |
| GEM-pBS | 4.49 | 5.74 | 8.73 | 22.01 | **9.01** | **18.89** | 4.27 | 10.12 | **2.01** | 2.32 |
| *2nd Order Stochastic Methods* | | | | | | | | | | |
| SOSaG | 4.47 | 8.00 | 7.88 | 25.55 | 11.71 | 20.70 | 4.18 | 11.10 | 2.52 | 3.23 |
| SOSaG-pBS | **3.96** | **5.28** | 8.68 | 24.61 | 12.99 | 20.50 | 4.22 | 11.24 | 2.08 | **2.17** |

the proposal stands for mutation, and the resampling act as a selection function. Evolutionary algorithms are inspired by biological models, and they have been largely shown to work in many machine learning problems (Wang et al., 2024; Novikov et al., 2025; Jiang et al., 2026). As such, the effectiveness of SMC samplers for diffusion models albeit the *statistical* degeneracy, may still be justified on these grounds. We have expanded further upon this discussion in Appendices A.5 and A.6.

## 4. Experiments

### 4.1. Benchmark PDEs

We first test our results on five common benchmark PDEs. Darcy flow (DF), inhomogeneous Helmholtz equation (IHE), non-bounded Navier–Stokes (NBNS), bounded Navier–Stokes (BNS) and the Poisson Equation. We have provided further details of these in Appendix B.1.1. Our experimental implementation is based on the open-source code released by Huang et al. (2024), which we adapted and extended for our specific experimental setting. We used the pretrained models they provided. As a baseline, we used the DiffusionPDE sampling method (Huang et al., 2024) with guidance (DiffPDE). We restricted our comparison to this as the original DiffPDE significantly outperformed other baselines such as PINNs, Deep-ONets, PINOs and FNOs; see Huang et al. (2024, Section 4).

We compare the baselines against three configurations of our SMC method, each using $N = 4$ particles and the proposal variant of SOSaG which is the equivalent of $N = 1$ in an SMC framework. Note that this is similar to DiffPDE but with stochastic instead of deterministic sampling. We use the GEM proposal with both the TDS and

the pBS framework, while the SOSaG proposal requires the pBS weighting. For both $u$ and $a$ our sparse observations consist of 500 pixels for each experiment. Figure 1 gives examples of these observations. The PDE snapshots each have $128 \times 128$ spatial grids which means we only observe $\approx 3.05\%$ of the PDE data information. All results were averaged over 30 independent runs.

### 4.2. Multiphysics PDE Systems with Observation Noise

Next, we test the methods on a multiphysics system of interacting PDEs. For this we chose a 2-species and 3-species Reaction-Diffusion (Kondo & Miura, 2010; Britton, 1986) (2SRD and 3SRD respectively), where with 2SRD we have to estimate the two scalar fields and two coefficient fields, one for each interacting PDE. For the 3SRD experiment, we aim to recover a solution field $(u, v, z)$ and coefficient field $(D_u, D_v, D_z)$ associated with each interacting PDE. Note that in this experiment, we are solving a *joint* inference problem, where we have observations on each coefficient and solution field, as opposed to the previous problems where we only have observations on either the coefficient or solution field. We have provided more details of this in Appendix B.1.2. We also tested both examples with varying levels of noise $\sigma_O$ added to the observations in order to test the ability of the sampling methods under potentially more realistic scenarios, where the observations may not be perfectly aligned with the underlying equations. Results are given in Table 2

## 5. Discussion

Table 1 shows the results across the PDE benchmarks. The tables give the relative error in percent compared to the error calculated by finite element methods (FEMs) which

*Table 2.* Per-channel errors (%) for Gray-Scott RD (2-species) and 3-Species RD across varying observation noise levels $\sigma_O$.

| Method | Gray-Scott RD (2-Species) | | | | 3-Species RD | | | | | |
|---|---|---|---|---|---|---|---|---|---|---|
| | $D_u$ | $D_v$ | $u$ | $v$ | $D_u$ | $D_v$ | $D_w$ | $u$ | $v$ | $w$ |
| $\sigma_O = 0.0$ | | | | | | | | | | |
| DiffPDE | 2.29 | 1.43 | 0.03 | 3.18 | 2.92 | 3.64 | 6.45 | **0.05** | **0.24** | 0.50 |
| GEM-TDS | 5.68 | 3.23 | **0.02** | 2.70 | 3.66 | 4.93 | 7.43 | 0.05 | 0.27 | 0.55 |
| GEM-pBS | **1.72** | **1.18** | 0.02 | 2.39 | **2.66** | 2.70 | 4.83 | 0.06 | 0.25 | **0.48** |
| SOSaG-pBS | 1.81 | 1.24 | 0.03 | **2.36** | 2.77 | **2.38** | **3.66** | 0.07 | 0.27 | 0.49 |
| $\sigma_O = 0.005$ | | | | | | | | | | |
| DiffPDE | 4.30 | 4.35 | 0.17 | 11.47 | 4.44 | 5.99 | 8.46 | 0.32 | 1.02 | 1.40 |
| GEM-TDS | 7.23 | 8.04 | **0.17** | 10.92 | 5.59 | 7.62 | 9.77 | **0.31** | 1.01 | 1.40 |
| GEM-pBS | 4.08 | 4.70 | 0.17 | 13.74 | **4.25** | 4.95 | 6.86 | 0.31 | **0.99** | **1.34** |
| SOSaG-pBS | **3.56** | **3.47** | 0.17 | **9.15** | 4.65 | **4.84** | **6.00** | 0.32 | 1.03 | 1.34 |
| $\sigma_O = 0.01$ | | | | | | | | | | |
| DiffPDE | 6.19 | 6.78 | **0.24** | 16.05 | **6.26** | 9.11 | 11.58 | **0.50** | **1.56** | 2.08 |
| GEM-TDS | 10.86 | 10.69 | 0.26 | 16.76 | 8.73 | 11.72 | 15.69 | 0.51 | 1.61 | 2.11 |
| GEM-pBS | 7.07 | 8.71 | 0.30 | 22.55 | 6.46 | **7.36** | 9.25 | 0.51 | 1.60 | **2.05** |
| SOSaG-pBS | **5.45** | **5.51** | 0.25 | **12.77** | 6.57 | 7.43 | **8.87** | 0.53 | 1.73 | 2.12 |
| $\sigma_O = 0.02$ | | | | | | | | | | |
| DiffPDE | **8.14** | 9.70 | **0.32** | 20.64 | **8.82** | 14.70 | 16.64 | **0.79** | **2.65** | **3.29** |
| GEM-TDS | 13.82 | 17.37 | 0.39 | 24.88 | 12.31 | 17.89 | 20.68 | 0.87 | 2.69 | 3.36 |
| GEM-pBS | 11.52 | 14.34 | 0.47 | 31.90 | 10.68 | 12.90 | 14.60 | 0.89 | 2.80 | 3.34 |
| SOSaG-pBS | 8.42 | **9.06** | 0.32 | **16.78** | 10.16 | **12.33** | **14.43** | 0.89 | 2.90 | 3.56 |

are used to generate the data and therefore we assume to be the ground truth, with the standard deviation given in the subscript. We notice that the stochastic methods generally tend to outperform the baselines in both the estimation of $u$ and $a$ with the pBS variants of SMC outperforming the TDS implementation. Figure 2 shows the reconstructed fields and the relative errors for a single run for the DF experiment. We can see that the relative error is visually lower for the SMC methods. We have provided single run plots for all experiments in Appendix C. Appendix A.5 gives results for the Darcy, Helmholtz and Poisson problems with varying tempering parameter $\rho$ values. We do notice that higher tempering values generally give better reconstruction error, which lends empirically validity to our pBS approach.

Tables 2 and 3 show the results for the 2SRD and 3SRD experiments, with the former showing the error for each coefficient and solution field and the latter displaying the average error across all fields. We can see that the performance degrades as the observation noise increases, an intuitive result. However, across nearly all results, the stochastic methods outperform the ODE ones. We also see that under higher noise levels, the SOSaG proposal tends to outperform

the GEM proposal. We hypothesize that under noisy observations, the second order correction present in the SOSaG proposal helps account for this and therefore produces superior results which may lead to it being a better choice of sampling method in real-world scenarios. For the 3SRD experiments, we observe that DiffPDE achieves good results on the solution fields. This may indicate that in noisy settings, second-order corrections have a greater effect on reconstruction accuracy.

## 6. Limitations and Further Work

Appendix C.2 gives the results for a single sample with the stochastic methods and shows that generally the stochastic proposals outperform the deterministic methods with an equal compute budget. However, we acknowledge the extra benefit in performance from the SMC methods incur an increased time cost compared to the baseline as the number of samples increases. An optimal implementation of the SMC methods would limit the extra wall clock time needed. The score model we call sequentially gets the score for each sample. This can be vectorized and will instantly speed up the algorithm.

*Table 3.* Combined results across all experiments and noise levels (%).

| Noise | Fields | DiffPDE | GEM-TDS | GEM-pBS | SOSaG-pBS |
|---|---|---|---|---|---|
| **Gray-Scott RD** | | | | | |
| $\sigma_O = 0$ | $D$ fields | 1.86 | 4.46 | **1.45** | 1.52 |
| | $u$ fields | 1.60 | 1.36 | 1.21 | **1.20** |
| $\sigma_O = 0.005$ | $D$ fields | 4.32 | 7.64 | 4.39 | **3.52** |
| | $u$ fields | 5.82 | 5.54 | 6.96 | **4.66** |
| $\sigma_O = 0.01$ | $D$ fields | 6.49 | 10.77 | 7.89 | **5.48** |
| | $u$ fields | 8.15 | 8.51 | 11.43 | **6.51** |
| $\sigma_O = 0.02$ | $D$ fields | 8.92 | 15.59 | 12.93 | **8.74** |
| | $u$ fields | 10.48 | 12.64 | 16.19 | **8.55** |
| **3-Species RD** | | | | | |
| $\sigma_O = 0$ | $D$ fields | 4.34 | 5.34 | 3.40 | **2.94** |
| | $u$ fields | 0.27 | 0.29 | **0.26** | 0.28 |
| $\sigma_O = 0.005$ | $D$ fields | 6.30 | 7.66 | 5.36 | **5.17** |
| | $u$ fields | 0.91 | 0.91 | **0.88** | 0.90 |
| $\sigma_O = 0.01$ | $D$ fields | 8.99 | 12.04 | 7.69 | **7.63** |
| | $u$ fields | **1.38** | 1.41 | 1.39 | 1.46 |
| $\sigma_O = 0.02$ | $D$ fields | 13.39 | 16.96 | 12.73 | **12.31** |
| | $u$ fields | **2.24** | 2.31 | 2.34 | 2.45 |

Although Appendix A.5 shows increased performance with higher tempering values, there is a drop off in performance as this is increased and in some cases a regression, meaning that the tuning of this hyperparameter is an open question as it cannot simply be set to be arbitrarily high. Also, as the optimal tempering parameter differs from experiment to experiment, it is likely that tuning of this is problem specific and therefore is still an open question.

The PDE residual intermediate likelihoods (and their gradients) can be computationally demanding to evaluate, especially for high-order and large PDE systems. It would be interesting to explore numerical acceleration techniques for the likelihood computation, and incorporate it within the SMC framework as an (possibly unbiased) estimation of the potential function. Another direction might be assuming a pre-trained PINN and use it as a distilled surrogate for computing the intermediate likelihoods.

## Acknowledgment

This work was financially supported by the Swedish Research Council (project no: 2024-05011), the Wallenberg AI, Autonomous Systems and Software Program (WASP) funded by the Knut and Alice Wallenberg (KAW) Foundation, and the Excellence Center at Linköping–Lund in Information Technology (ELLIIT). Computations were enabled by the Berzelius resource at the National Supercomputer Centre, provided by the Knut and Alice Wallenberg Foundation.

## Impact Statement

This paper presents work whose goal is to advance the field of Machine Learning. There are many potential societal consequences of our work, none which we feel must be specifically highlighted here.

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

# A. Sequential Monte Carlo Derivations

## A.1. Sequential Monte Carlo Pseudocode

---

**Algorithm 1** Sequential Monte Carlo (SMC)

---

**Require:** Number of particles $N$, time steps $K$; proposals $M_K(x_K), M_k(x_{k-1} \mid x_k)$; incremental weight functions $G_K(x_K), G_k(x_{k-1}, x_k)$; resampling threshold $N_{eff} \in (0, 1]$.

1: **Initialization:**
2: **for** $n = 1 \to N$ **do**
3: $\quad x_K^{(i)} \sim M_K(x_K^{(i)})$
4: $\quad w_K^{(i)} = G_K(x_K^{(i)})$
5: **end for**
6: **for** $k = K \to 1$ **do**
7: $\quad$ **for** $i = 1 \to N$ **do**
8: $\quad\quad$ **Propagate:** $x_{k-1}^{(i)} \sim M_k(x_{k-1}^{(i)} \mid x_k^{(i)})$
9: $\quad\quad$ **Weight update:** $\tilde{w}_{k-1}^{(i)} = w_k^{(i)} G_k(x_{k-1}^{(i)}, x_k^{(i)})$
10: $\quad$ **end for**
11: $\quad$ **Normalize:** $w_{k-1}^{(i)} = \tilde{w}_{k-1}^{(i)} \Big/ \sum_{j=1}^N \tilde{w}_{k-1}^{(j)}$, for all $i$
12: $\quad$ **Compute effective sample size:** $\text{ESS}_k = \frac{1}{\sum_{i=1}^N (w_{k-1}^{(i)})^2}$
13: $\quad$ **if** $\text{ESS}_k \leq \tau$ **then**
14: $\quad\quad$ **Resampling (Multinomial):**
15: $\quad\quad$ **for** $i = 1 \to N$ **do**
16: $\quad\quad\quad a^{(i)} \sim \text{Cat}(w_{k-1}^{(1)}, \dots, w_{k-1}^{(N)})$
17: $\quad\quad\quad x_{k-1}^{(i)} = x_{k-1}^{(a^{(i)})}$
18: $\quad\quad\quad w_{k-1}^{(i)} = \frac{1}{N}$
19: $\quad\quad$ **end for**
20: $\quad$ **end if**
21: **end for**
22: **Return:** $\{(x_0^{(i)}, w_0^{(i)})\}_{i=1}^N$

---

## A.2. EM Proposal

The FK formulation used to derive the targets is provided in equation 9 but we have restated it here for convenience purposes.

$$\nu(x_{0:K}) = \frac{1}{\mathcal{L}} G_K(x_K) M_K(x_K) \prod_{j=1}^k G_{j-1}(x_j, x_{j-1}) M_{j-1}(x_{j-1} \mid x_j) \tag{21}$$

Where $\mathcal{L}$ is the normalisation constant. Using

$$M_K(x_K \mid x_K) = p_K(x_K), \qquad\qquad M_{j-1}(x_{j-1} \mid x_j) = \tilde{p}_\theta(x_{j-1} \mid x_j, y),$$

$$G_K(x_K) = \tilde{p}_\theta(y \mid x_K), \qquad\qquad G_{j-1}(x_j, x_{j-1}) = \frac{\tilde{p}_\theta(y \mid x_{j-1})}{\tilde{p}_\theta(y \mid x_j)} \frac{p_\theta(x_{j-1} \mid x_k)}{\tilde{p}_\theta(x_{j-1} \mid x_j, y)}.$$

Substituting these into equation 21:

$$\nu(x_{s:K}) \propto \tilde{p}_\theta(y \mid x_K) p_K(x_K) \prod_{j=s}^k \frac{\tilde{p}_\theta(y \mid x_{j-1})}{\tilde{p}_\theta(y \mid x_j)} \frac{p_\theta(x_{j-1} \mid x_j)}{\tilde{p}_\theta(x_{j-1} \mid x_j, y)} \tilde{p}_\theta(x_{j-1} \mid x_j, y), \tag{22}$$

$$= \tilde{p}_\theta(y \mid x_K) p_K(x_K) \prod_{j=s}^k p_\theta(x_{j-1} \mid x_k), \tag{23}$$

$$= \tilde{p}_\theta(y \mid x_{s-1}) p_\theta(x_{s-1:K}), \tag{24}$$

$$= p(x_{s-1:K}, y). \tag{25}$$

And therefore:

$$\nu(x_{s:K}) = p(x_{s-1:K} \mid y), \tag{26}$$
$$\nu(x_{0:K}) = p(x_{0:K} \mid y). \tag{27}$$

## A.3. Pseudo-Bootstrap Formulation

For the pBS formulation of the FK process, we use the following:

$$M_K(x_K) = p_K(x_K), \qquad\qquad M_{j-1}(x_{j-1} \mid x_j) = \tilde{p}_\theta(x_{j-1} \mid x_j, y),$$

$$G_K(x_K) = \tilde{p}_\theta(y \mid x_K), \qquad\qquad G_{j-1}(x_j, x_{j-1}) = \frac{\tilde{p}_\theta(y \mid x_{j-1})}{\tilde{p}_\theta(y \mid x_j)}.$$

Substituting these into equation 21:

$$\nu(x_{s:K}) \propto \tilde{p}_\theta(y \mid x_K) p_K(x_K) \prod_{j=s}^{k} \frac{\tilde{p}_\theta(y \mid x_{j-1})}{\tilde{p}_\theta(y \mid x_j)} \tilde{p}_\theta(x_{j-1} \mid x_j, y), \tag{28}$$

$$= \tilde{p}_\theta(x_{s-1:K}, y) \tilde{p}_\theta(y \mid x_{s-1}). \tag{29}$$

## A.4. Pseudocode

### A.4.1. GEM PROPOSAL PSEUDOCODE

---
**Algorithm 2** GEM Algorithm
---
**Require:** $D_\theta(x; \sigma)$, $\sigma_k, \sigma_{k-1}, x_k, \alpha$
1: Sample $\epsilon_k \sim \mathcal{N}(0, I)$
2: $d_k \leftarrow \dfrac{x_k - D_\theta(x_k, \sigma_k)}{\sigma_k}$
3: $x_{k-1} \leftarrow x_k + (\sigma_{k-1}^2 - \sigma_k^2) d_k + \sqrt{\sigma_{k-1}^2 - \sigma_k^2} \epsilon_k$
4: $x_{k-1} \leftarrow x_{k-1} - (\sigma_{k-1}^2 - \sigma_k^2) \nabla_{x_k} \log \tilde{p}_\theta(y \mid x_k)$
5: **return** $x_{k-1}$

---

### A.4.2. SOSAG PROPOSAL PSEUDOCODE

---
**Algorithm 3** SOSaG Proposal
---
**Require:** $D_\theta(x, \sigma)$, $\sigma_k, \sigma_{k-1}, x_k, \gamma_k$
1: Sample $\epsilon_k \sim \mathcal{N}(0, I)$
2: $\hat{\sigma}_k \leftarrow \sigma_k + \gamma_k \sigma_k$
3: $\hat{x}_k \leftarrow x_k + \sqrt{\hat{\sigma}_k^2 - \sigma_k^2} \epsilon_k$
4: $d_k \leftarrow \dfrac{\hat{x}_k - D_\theta(\hat{x}_k, \hat{\sigma}_k)}{\hat{\sigma}_k}$
5: $x_{k-1} \leftarrow \hat{x}_k + (\sigma_{k-1} - \hat{\sigma}_k) d_k$
6: **if** $\sigma_k \neq 0$ **then**
7: $\quad d_{k-1} \leftarrow \dfrac{x_{k-1} - D_\theta(x_{k-1}, \sigma_{k-1})}{\sigma_{k-1}}$
8: $\quad x_{k-1} \leftarrow \hat{x}_k + (\sigma_{k-1} - \hat{\sigma}_k)\left(\frac{1}{2}d_k + \frac{1}{2}d_{k-1}\right)$
9: **end if**
10: $x_{k-1} \leftarrow x_{k-1} - (\sigma_{k-1}^2 - \sigma_k^2) \nabla_{\hat{x}_k} \log \tilde{p}_\theta(y \mid \hat{x}_k)$
11: **return** $x_{k-1}$

---

### A.5. Clarification of posterior exactness and empirical performance

We clarify a concern raised by the area chair during the review process on whether pBS trades between the posterior exactness and empirical performance.

PDE solutions are intrinsically deterministic whereas diffusion models are stochastic. Solving PDEs via diffusion models essentially applies Bayesian inference to a frequentist problem, introducing a large degree of freedom in defining the target posterior distribution. Most state-of-the-art diffusion models define the target as

$$\overbrace{\pi(u \mid u_{\mathrm{obs}}, a)}^{\text{Diffusion PDE target}} \propto \overbrace{p(u_{\mathrm{obs}} \mid u)}^{\text{Partial observations from data}} \times \overbrace{r_a(u)}^{\text{PDE constraints}} \times \overbrace{p(u)}^{\text{Diffusion prior}}, \tag{30}$$

by incorporating the partial observations likelihood, PDE constraints likelihood, and a pre-trained diffusion prior. Then, samplers (e.g., SMC) are built aiming to statistically exactly sample from this target. However, this ansatz ignores the fact that both the diffusion prior and the likelihoods contain irreducible errors. Sampling exactly from this target does not guarantee a valid solution from the PDE, and the errors propagating to the samples is not controlled. This motivates us introducing pseudo-bootstrap (pBS), modifying from Equation (30) with additional tempering exponent terms on the prior and likelihood to calibrate their respective error contributions.

Table 4 give the results on the Darcy, Helmholtz and Poisson forward and inverse experiments when varying the tempering parameter $\rho$ in Equation (20). We can see from this table that higher values of $\rho$ result in lower relative errors. Consequently, pBS does not trade between the posterior exactness and empirical performance. Instead, pBS produces statistically exact samples from an alternative target distribution that is calibrated against the standard target in Equation (30). Empirically, samples drawn from this new target demonstrate superior accuracy compared to those obtained from the original distribution.

There is also previous precedence for exploring tuning this tempering parameter. Wu et al. (2023) explored the use of changing this parameter which they referred to as the *twist scale* and found increasing it produced better results for class conditional generation for the MNIST dataset (Appendix D.2.1).

*Table 4.* Results for Darcy, Helmholtz and Poisson experiments with differing values of $\rho$. Results are averaged over 20 runs.

| Method | $\rho=5$ | $\rho=10$ | $\rho=50$ | $\rho=100$ | $\rho=500$ | $\rho=1000$ |
|---|---|---|---|---|---|---|
| | | | *Darcy – Forward* | | | |
| GEM-pBS | 6.19% | 4.46% | 3.58% | 4.50% | 3.91% | 4.07% |
| SOSaG-pBS | 4.52% | 3.22% | 3.55% | 4.20% | **3.19%** | 3.42% |
| | | | *Darcy – Inverse* | | | |
| GEM-pBS | 6.05% | 5.60% | 5.32% | 4.69% | 4.89% | 5.08% |
| SOSaG-pBS | 4.86% | 4.90% | 4.69% | **4.55%** | 4.77% | 4.56% |
| | | | *Helmholtz – Forward* | | | |
| GEM-pBS | 13.37% | 10.72% | 9.29% | 8.90% | 7.85% | 10.27% |
| SOSaG-pBS | 11.16% | 10.95% | **7.69%** | 9.20% | 8.40% | 8.67% |
| | | | *Helmholtz – Inverse* | | | |
| GEM-pBS | 18.55% | 18.64% | 18.63% | 18.45% | 17.68% | 18.08% |
| SOSaG-pBS | 20.12% | 19.80% | 19.18% | 19.30% | 18.17% | **17.57%** |
| | | | *Poisson – Forward* | | | |
| GEM-pBS | 6.58% | 6.79% | 6.18% | 7.05% | 6.09% | 6.61% |
| SOSaG-pBS | 6.55% | 8.10% | 7.89% | 8.54% | **6.03%** | 6.86% |
| | | | *Poisson – Inverse* | | | |
| GEM-pBS | 22.09% | 22.18% | 21.96% | 21.12% | 20.65% | 20.34% |
| SOSaG-pBS | 24.13% | 23.92% | 23.76% | 22.70% | 21.31% | **20.17%** |

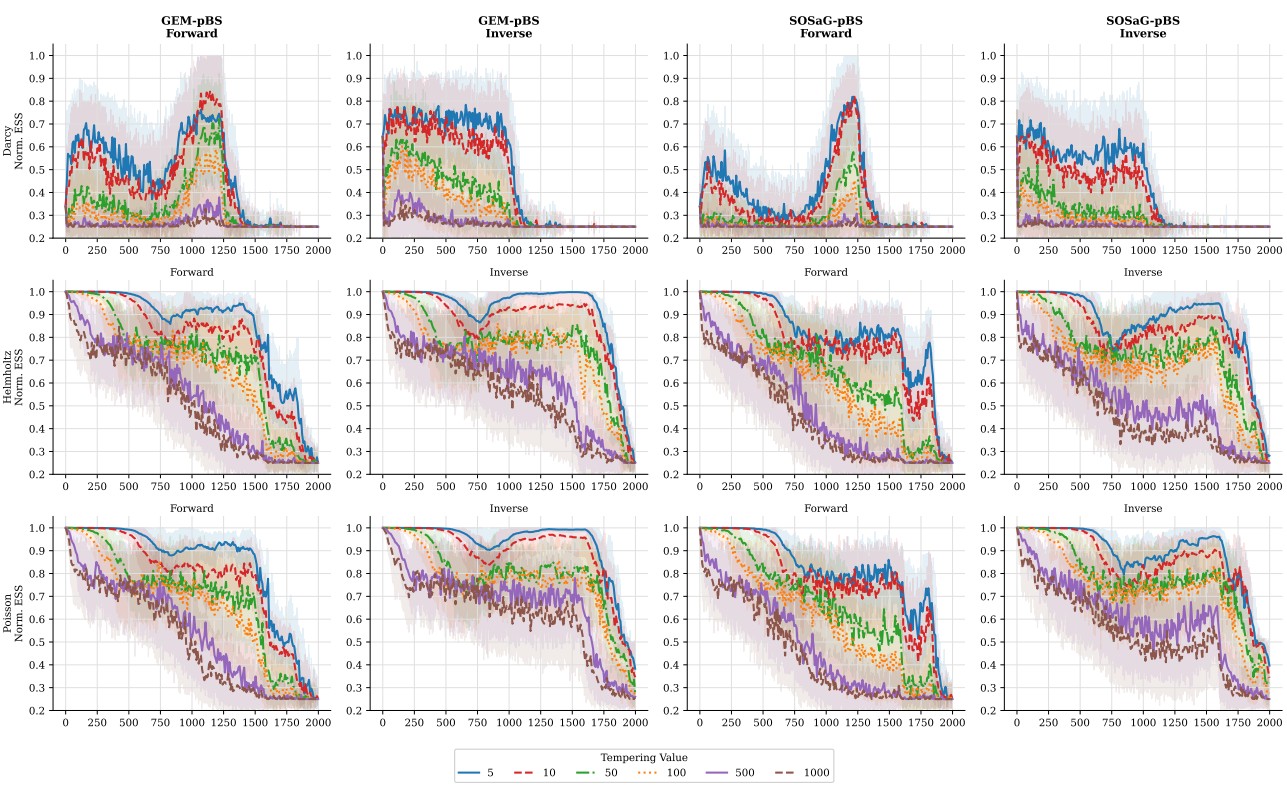

*Figure 3.* Normalised effective sample size during the denoising process with different particle size.

## A.6. Tempered ESS and Genetic Algorithms

Increasing $\rho$ has the effect of collapsing the ESS of the SMC process, as shown in Figure 3. As discussed in Section 3.5, this collapsing ESS reveals a duality between SMC algorithms and evolutionary algorithms (Bäck & Schwefel, 1993; Yu & Gen, 2010, EAs): both can be cast as instances of a Feynman–Kac model (Del Moral, 2004), and the correspondence follows by inspection of terms.

At values of $\rho \to \infty$, the ESS converges to a single particle and the sampler behaves like the $1 + \lambda$ EA (Yu & Gen, 2010; Bäck & Schwefel, 1993). This algorithm starts with a single parent which spawns $N$ children; the best-performing child is then used to spawn $N$ children at the next iteration, and the process repeats until convergence. It has been observed previously that when using SMC to sample from diffusion models, the ESS collapses towards the end of the sampling run, likely because the noise variance approaches $0$ (the noise schedule is often linear, going from $0.2 \to 0.01$). Therefore, even when low ESS is induced from the start by setting the variance parameter to be small, many existing SMC methods will experience this collapse in the latter stages of sampling. When the tempering parameter $\rho$ is large and the target is dominated by the likelihood, the EA approximately corresponds to maximum likelihood estimation (MLE); when $\rho$ is small and the prior plays a larger role, it resembles maximum a posteriori (MAP) estimation, making the algorithm analogous to a Bayesian EA. This can be illustrated as follows.

**1. Mutation $\longleftrightarrow$ Proposal.** In an EA, each individual $x_j$ in the population is stochastically perturbed to produce an offspring $x_{j-1}^{(i)}$. In pBS, each particle is propagated via the guided sampler $x_{j-1}^{(i)} \sim M_{j-1}(\cdot \mid x_j) = \tilde{p}_\theta(x_{j-1} \mid x_j, y)$. Both operations apply a stochastic perturbation to each member of the population independently, conditional on its current state.

**2. Fitness evaluation $\longleftrightarrow$ Weighting.** In an EA, each offspring is assigned a scalar fitness value. In pBS, each particle receives an incremental weight

$$G_{j-1}(x_j, x_{j-1}) = \frac{\tilde{p}_\theta(y \mid x_{j-1})}{\tilde{p}_\theta(y \mid x_j)}.$$

In the deterministic (PDE) setting, the approximate likelihood $\tilde{p}_\theta(y \mid x_{j-1})$ measures how well the denoised reconstruction $\delta_\theta(x_{j-1}, \sigma_{j-1})$ satisfies the observations and the PDE residual, and therefore plays the role of a fitness function: particles whose reconstructions better satisfy the physics and the data receive higher weight.

**3. Selection $\longleftrightarrow$ Resampling.** In an EA, individuals are selected for the next generation with probability proportional to their fitness. In SMC, when the effective sample size drops below $N_{\text{eff}}$, particles are resampled with probability proportional to their normalised weights. Both operations duplicate high-fitness/high-weight individuals and discard low-fitness/low-weight ones, concentrating the population around promising regions of the search space.

# B. Supplementary Experimental Design Material

## B.1. PDE Equation Details

### B.1.1. BENCHMARK PDE DETAILS

**Darcy Flow:** In Darcy flow, we solve for the pressure field $u(c)$ given the permeability field $a(c)$ of a specific medium $-\nabla \cdot (a(c)\nabla u(c)) = s(c)$

$$
\begin{aligned}
-\nabla \cdot (a(c)\nabla u(c)) &= s(c), \quad c \in \Omega \\
u(c) &= 0, \qquad c \in \partial\Omega,
\end{aligned}
\tag{31}
$$

and $s(c)$ is a source term such as a forcing function. For these experiments, we set $s(c) = 1$.

**Inhomogeneous Helmholtz/Poisson Equation:** In the inhomogeneous Helmholtz equation we solve for the wave field $u(c)$ given a forcing term $a(c)$ which describes wave propagation:

$$
\begin{aligned}
\nabla^2 u(c) + k^2 u(c) &= a(c), \quad c \in \Omega \\
u(c) &= 0, \qquad c \in \partial\Omega.
\end{aligned}
\tag{32}
$$

$k$ is a constant (wavenumber) and $a(c)$ is a piecewise constant function. Note that when $k = 0$, this reduces to the Poisson equation:

$$
\begin{aligned}
\nabla^2 u(c) &= a(c), \quad c \in \Omega \\
u(c) &= 0, \qquad c \in \partial\Omega.
\end{aligned}
\tag{33}
$$

**Non-Bounded Incompressible Navier–Stokes:** The following describes the vorticity formulation of the non-bounded Navier-Stokes equation:

$$
\begin{aligned}
\frac{\partial w(c,\tau)}{\partial \tau} + v(c,\tau) \cdot \nabla w(c,\tau) &= \nu \nabla w(c,\tau) + q(c), \quad c \in \Omega, \qquad \tau \in [0,\mathcal{T}], \\
\nabla \cdot v(c,\tau) &= 0, \qquad\qquad\qquad\qquad c \in \Omega, \qquad \tau \in [0,\mathcal{T}],
\end{aligned}
\tag{34}
$$

where $w = \nabla \times v$ is the vorticity. $v(c,\tau)$ is the velocity at $c$ at time $\tau$. $q(c)$ again is a forcing term but represented as a field. We follow the same set up as (Huang et al., 2024), setting $\nu = 1 \times 10^{-3}$ and learning the joint distribution of $w_0$ and $w_\mathcal{T}$. The dynamics are simulated over $\mathcal{T} = 10$ time steps which is equivalent to one second. The guidance is simplified to $f = \nabla \cdot w(c,\tau)$ as we cannot compute the PDE loss from 34 with our model outputs.

**Bounded Incompressible Navier–Stokes:** We consider the bounded 2D Navier-Stokes equation:

$$
\begin{aligned}
\frac{\partial v(c,\tau)}{\partial \tau} + v(c,\tau) \cdot \nabla v(c,\tau) + \frac{1}{\rho}\nabla p &= \nu \nabla^2 v(c,\tau), \quad c \in \Omega, \qquad \tau \in [0,\mathcal{T}], \\
\nabla \cdot v(c,\tau) &= 0, \qquad\qquad\quad c \in \Omega, \qquad \tau \in [0,\mathcal{T}],
\end{aligned}
\tag{35}
$$

where $\rho = 1$ and $\nu = 1 \times 10^{-3}$. Again following the data generation process from (Huang et al., 2024), 2D cylinders of random radius and points are generated inside the grid. We learn the joint distribution of $v_0, v_\mathcal{T}$ where $\mathcal{T} = 4$ and this simulates 0,4 seconds. Similarly to the Nonbounded case, we use $f = \nabla \cdot v(c,\tau)$.

### B.1.2. REACTION–DIFFUSION EQUATIONS (2-SPECIES AND 3-SPECIES)

We now explain the 2SRD and 3SRD PDE equations with periodic boundary conditions.

**Two-species reaction–diffusion equation (Gray–Scott):** We consider a two-species reaction–diffusion (RD) system with solution fields $u(c,\tau)$ and $v(c,\tau)$ describing the concentrations of two interacting species. The dynamics are given by the Gray–Scott model:

$$
\begin{aligned}
\frac{\partial u(c,\tau)}{\partial \tau} &= D_u \Delta u(c,\tau) - u(c,\tau)v(c,\tau)^2 + F\big(1 - u(c,\tau)\big), \quad c \in \Omega, \qquad \tau \in [0,\mathcal{T}], \\
\frac{\partial v(c,\tau)}{\partial \tau} &= D_v \Delta v(c,\tau) + u(c,\tau)v(c,\tau)^2 - (F + r)v(c,\tau), \quad c \in \Omega, \qquad \tau \in [0,\mathcal{T}].
\end{aligned}
\tag{36}
$$

Here $D_u$ and $D_v$ are diffusivities, $F$ is a feed rate and $r$ is a removal rate. The nonlinear coupling term $uv^2$ models autocatalytic production of $v$ at the expense of $u$. In our experiments we set $F = 0.035$ and $r = 0.060$. We simulate the system over $\mathcal{T} = 1$ second, recording 10 evenly spaced time steps, and generate trajectories using FEM with time step $1 \times 10^{-4}$. Initial conditions are sampled as a perturbed uniform state with a centered square perturbation: $u_0(c) \approx 1$ and $v_0(c) \approx 0$ throughout the domain, except on a centered patch where $u_0(c) \approx 0.5$ and $v_0(c) \approx 0.25$, with small additive Gaussian noise. We learn the joint distribution of the initial and final states $(u_0, v_0)$ and $(u_\mathcal{T}, v_\mathcal{T})$.

We define the PDE guidance function as the reaction–diffusion residual $f = (f_u, f_v)$, where $f_u = \partial_\tau u - D_u \Delta u + uv^2 - F(1-u)$ and $f_v = \partial_\tau v - D_v \Delta v - uv^2 + (F+k)v$.

**Three-species reaction–diffusion equation:** We also consider a three-species reaction–diffusion system with solution fields $u(c, \tau)$, $v(c, \tau)$, and $z(c, \tau)$ governed by diffusion and competitive interactions:

$$
\begin{aligned}
\frac{\partial u(c,\tau)}{\partial \tau} &= \nabla \cdot \big(D_u \nabla u(c,\tau)\big) + u(c,\tau)\Big(1 - u(c,\tau) - a_{12}v(c,\tau) - a_{13}z(c,\tau)\Big), \quad c \in \Omega, \qquad \tau \in [0, \mathcal{T}], \\
\frac{\partial v(c,\tau)}{\partial \tau} &= \nabla \cdot \big(D_v \nabla v(c,\tau)\big) + v(c,\tau)\Big(1 - v(c,\tau) - a_{21}u(c,\tau) - a_{23}z(c,\tau)\Big), \quad c \in \Omega, \qquad \tau \in [0, \mathcal{T}], \qquad (37) \\
\frac{\partial z(c,\tau)}{\partial \tau} &= \nabla \cdot \big(D_z \nabla z(c,\tau)\big) + z(c,\tau)\Big(1 - z(c,\tau) - a_{31}u(c,\tau) - a_{32}v(c,\tau)\Big), \quad c \in \Omega, \qquad \tau \in [0, \mathcal{T}],
\end{aligned}
$$

Here $D_u$, $D_v$, and $D_z$ denote diffusion coefficients. Here $a_{ij} \geq 0$ are competition coefficients quantifying how strongly species $j$ inhibits species $i$, and we collect them in a coupling matrix $A = (a_{ij}) \in \mathbb{R}^{3 \times 3}$ (with $a_{11} = a_{22} = a_{33} = 0$). In our data generation procedure, we sample the dominant cyclic interaction terms $a_{12}, a_{23}, a_{31} \sim \mathcal{U}(1.2, 2.0)$ and the remaining cross-interaction terms $a_{13}, a_{21}, a_{32} \sim \mathcal{U}(0.3, 0.8)$. $u_0(c) \approx 0.6$, $v_0(c) \approx 0.2$, and $z_0(c) \approx 0.2$ throughout the domain, except on three localized patches near the domain center where $u_0(c) \approx 0.9$, $v_0(c) \approx 0.7$, and $z_0(c) \approx 0.7$, respectively, with small additive Gaussian noise. We learn the joint distribution of the initial and final states $(u_0, v_0, z_0)$ and $(u_\mathcal{T}, v_\mathcal{T}, z_\mathcal{T})$.

Similarly, we define the PDE guidance function as the reaction–diffusion residual $f = (f_u, f_v, f_z)$, where $f_u = \partial_\tau u - D_u \Delta u - u(1 - u - a_{12}v - a_{13}z)$, $f_v = \partial_\tau v - D_v \Delta v - v(1 - v - a_{21}u - a_{23}z)$, and $f_z = \partial_\tau z - D_z \Delta z - z(1 - z - a_{31}u - a_{32}v)$.

## B.2. Model Training and Evaluation

For the benchmark PDEs we used the pretrained model parameters from Huang et al. (2024). The model is the modified UNet (Ronneberger et al., 2015) implementation used in Song et al. (2021) with each model trained on 50000 training images for each experiment. We refer the user to Huang et al. (2024) for further details on this.

For the multiphysics equations, we generated our own data using FEM to train the model architecture. For both the 2 species and 3 species variants, we generated 10000 training images and trained the model by doing 100,000 gradient passes over the dataset. A single A100 NVIDIA GPU was used for training.

For the evaluation, we generated 1000 test images and randomly chose a trajectory and then evaluated on the midpoint. We repeated this 20 times for each experiment and method, keeping the same image and known observations but with a different starting seed for particle initialization.

It is worth noting that we do not believe that our model was optimized with a scheme that leads to the optimal parameters. When training was terminated, the loss was still decreasing, albeit slowly. Therefore we think we all models could have superior results by optimizing this process. However, we felt that this task was outside the purview of this work as all methods including baselines still obtained good results. We also believe that there are potentially better architectures in the literature that would work better as a model prior than what we have chosen. The Song et al. (2021) was used for convenience and as that is what our baselines comparisons used for the pretrained models.

We found during some of our training runs that increasing $K$ improved all results across every sampling method, which is expected but the result was more pronounced for the stochastic sampling methods. We decided to use $K = 2000$ for each experiment and method, however we feel that experimenting with this parameter would be fruitful for more optimized results in future work. This obviously comes at an increased wall-clock time cost.

Table 5 gives the guidance parameters used for the forward, inverse and joint problems. It is worth noting that for the joint inference problems, a strong weighting on $a$ can cause a negative impact on the estimation of $u$ and vice versa. Therefore in

the joint inference scenario, guidance parameters have been tuned to minimise the overall residual error across the coefficient and solution fields.

*Table 5.* The weights assigned to the PDE loss and the observation loss vary depending on whether the observations pertain to the coefficients (or initial states) $a$ or to the solutions (or final states) $u$.

| | Benchmarks | | | | | Reaction Diffusion | |
|---|---|---|---|---|---|---|---|
| | Darcy | Poisson | Helmholtz | NS (Unbounded) | NS (Bounded) | 2SRD | 3SRD |
| $\gamma$ | 5 | 0.8 | 0.8 | 0.5 | 1 | 7 | 5 |
| $\beta$ | 5000 | 40 | 60 | 0.5 | 1 | 20 | 30 |
| $\omega$ | 100 | 1 | 1 | 1 | 1 | 2 | 1 |

# C. Further Results

## C.1. ESS Graphs

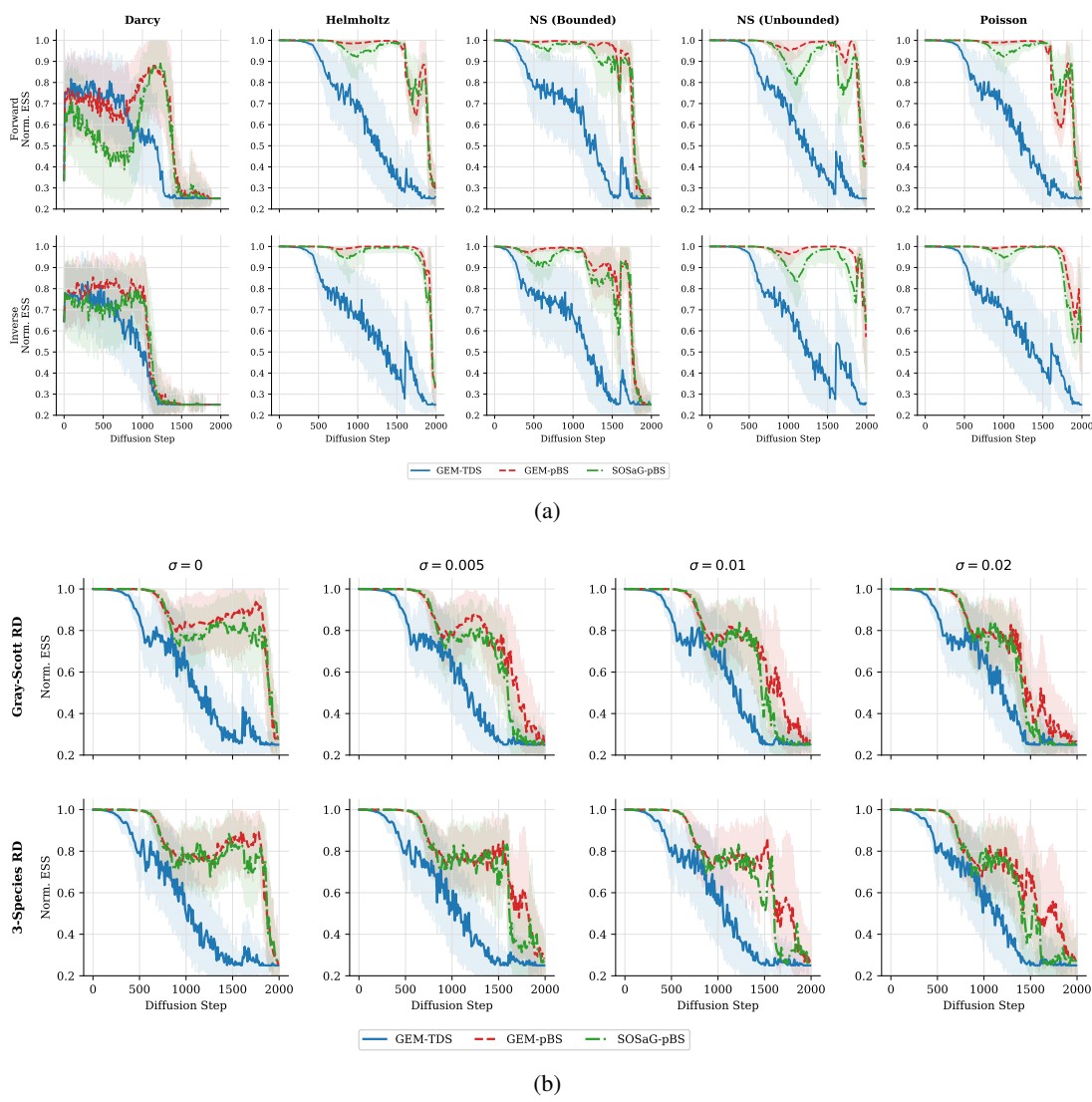

*Figure 4.* ESS plots for the Benchmark and RD experiments.

## C.2. Equal Computational Expense

One valid criticism of our algorithm is that we incur an extra computational cost when using the SMC methods compared to a single particle method such as DiffPDE. Table 6 give results on the benchmark datasets, but with the SMC framework removed. We still find that generally the stochastic methods produce better results than DiffPDE, and SMC often augments this advantage.

*Table 6.* Comparison of different models on five PDE problems (in $L_2$ relative error, except in the Darcy inverse problem where error rate is reported).

| Method | Darcy | | Poisson | | Helmholtz | | NS | | NS (BCs) | |
|---|---|---|---|---|---|---|---|---|---|---|
| | **Fwd** | **Inv** | **Fwd** | **Inv** | **Fwd** | **Inv** | **Fwd** | **Inv** | **Fwd** | **Inv** |
| DiffPDE | 5.58 | 8.31 | 8.67 | **19.88** | 17.49 | **19.14** | 4.27 | **9.34** | 2.78 | 3.43 |
| GEM | 5.28 | **7.34** | **6.53** | 22.28 | **11.22** | 19.30 | **4.16** | 10.10 | **2.31** | **2.36** |
| SOSaG | **4.47** | 8.00 | 7.88 | 25.55 | 11.71 | 20.70 | 4.18 | 11.10 | 2.52 | 3.23 |

## C.3. Error and Reconstruction Plots

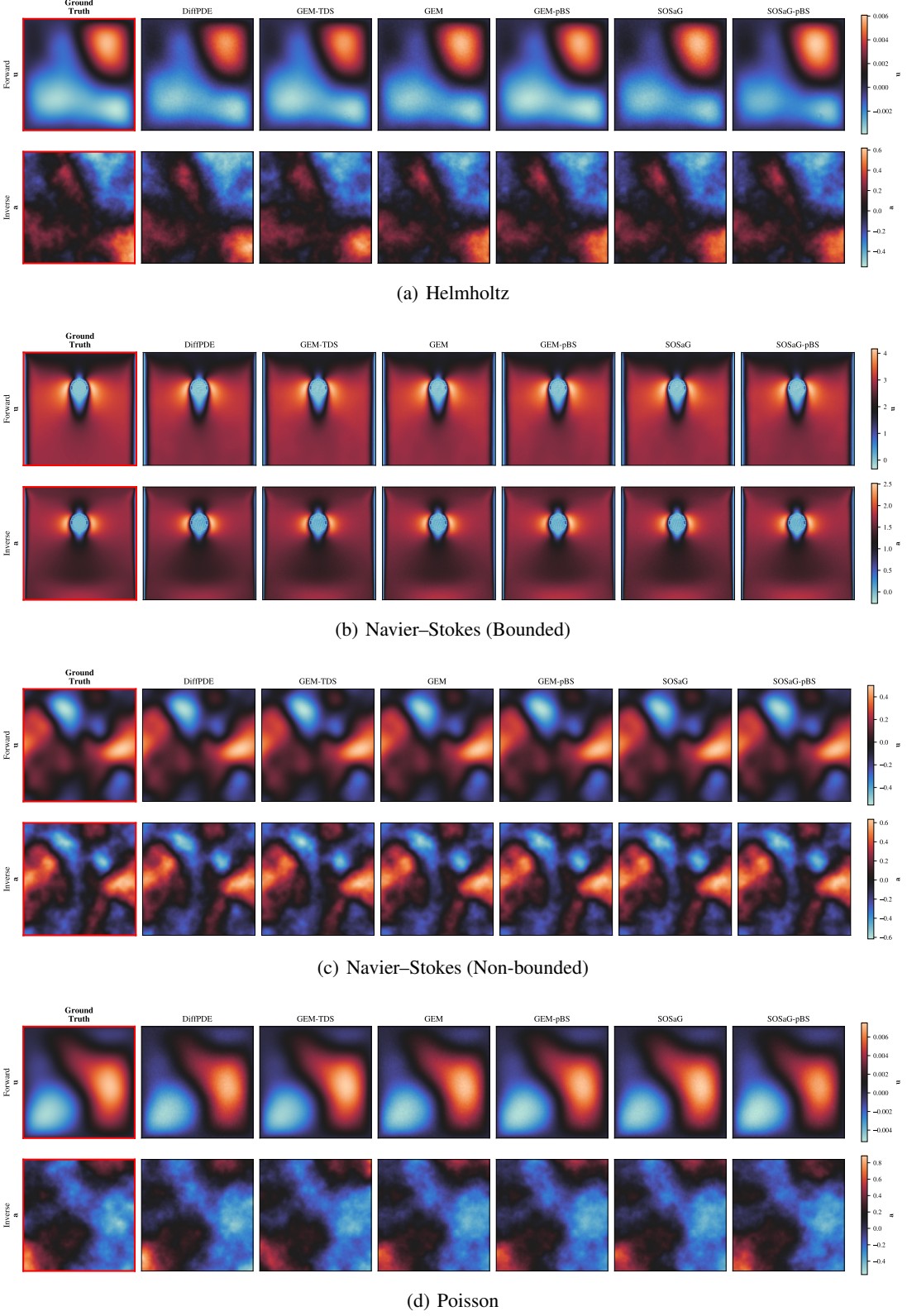

*Figure 5.* Reconstruction comparison for the benchmark PDE experiments. The top row shows the contour plots of the reconstruction of $a$ while the bottom row shows the reconstruction of $u$. The final two plots on the left show the ground truth of the respective fields. The figures show a single example run of each method.

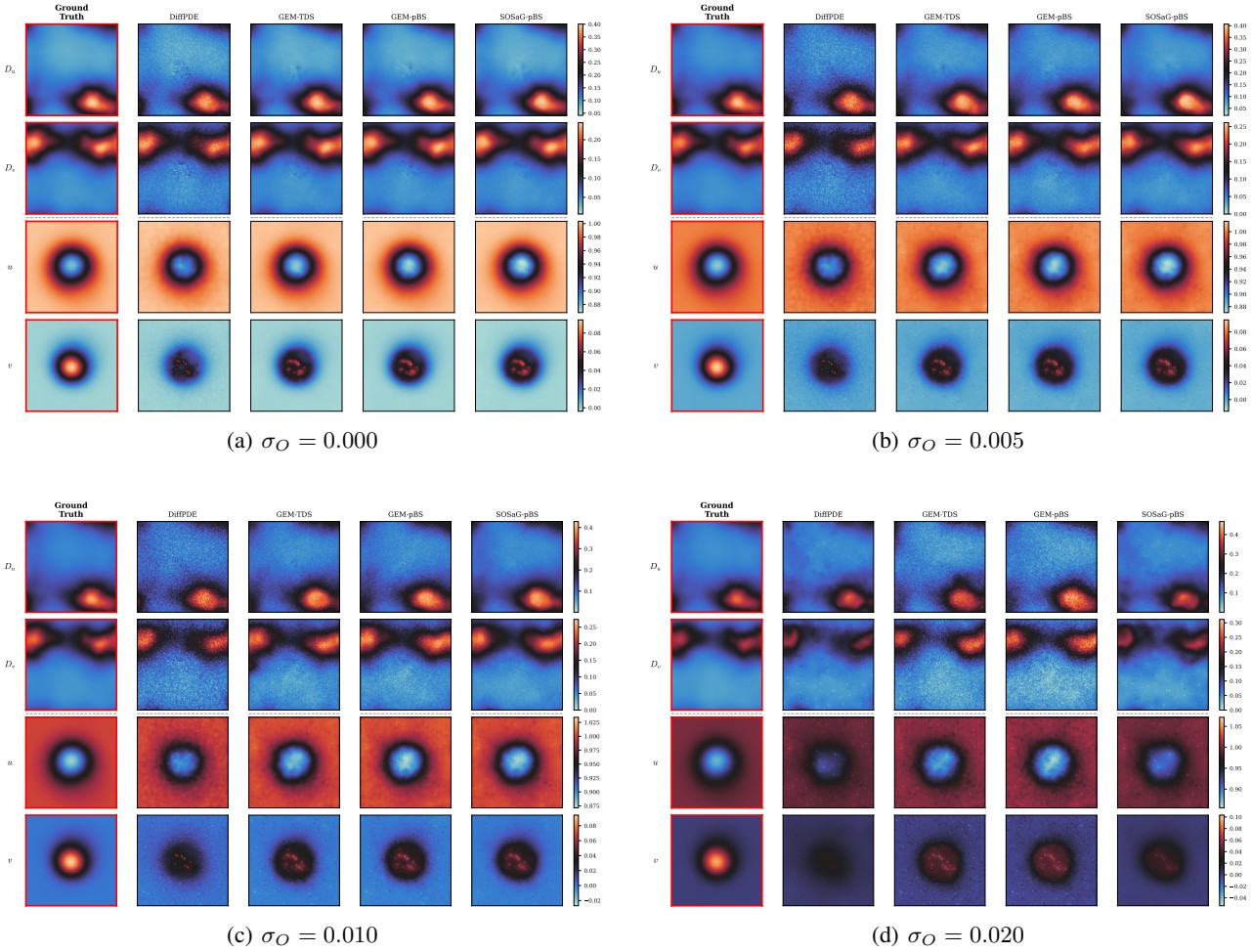

*Figure 6.* Reconstruction comparison for all 2SRD experiments across varying noise levels. The left most column show the ground truth contours. The figures show a single example run of each method.

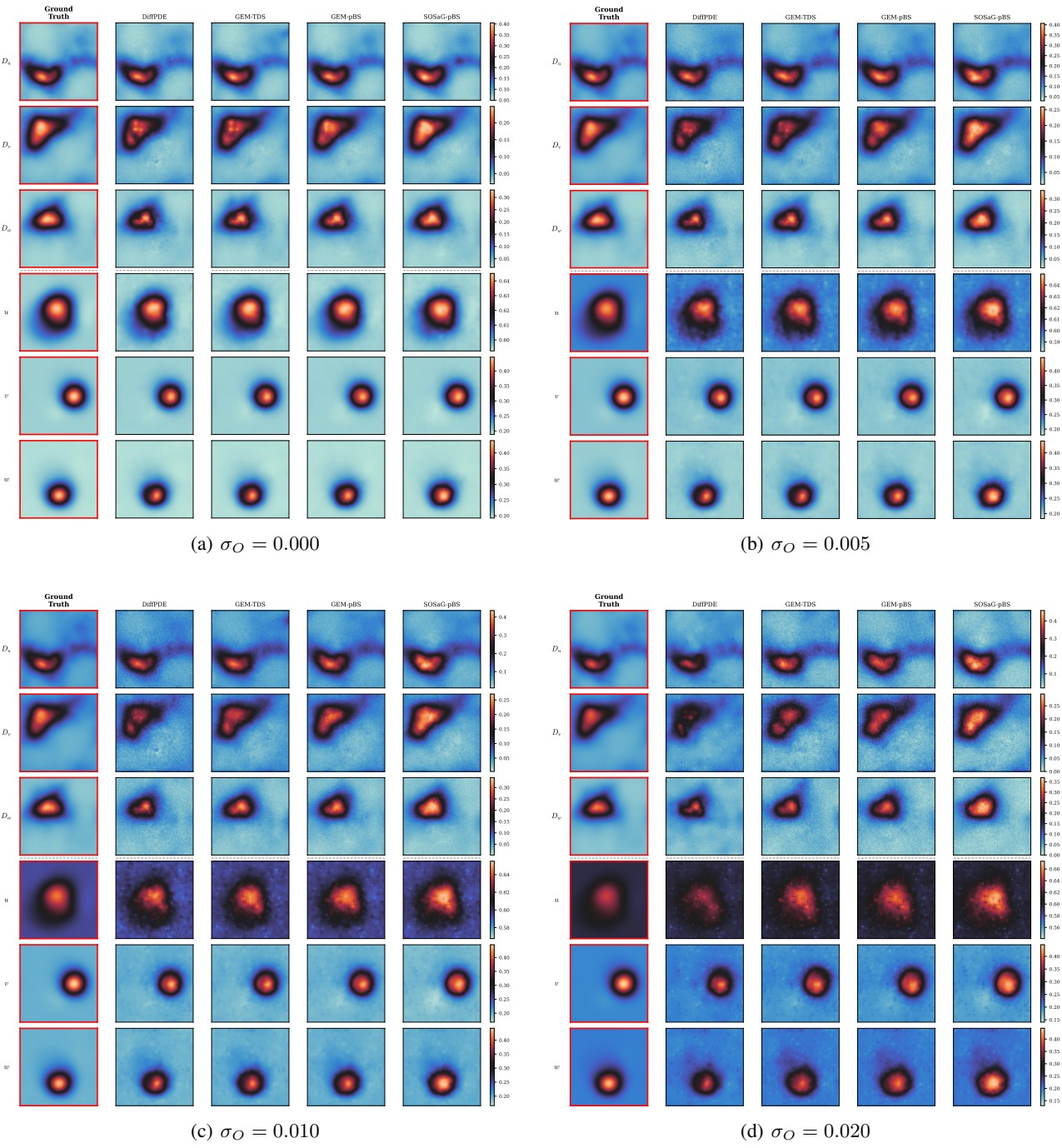

*Figure 7.* Reconstruction comparison for all 3SRD experiments across varying noise levels. The left most column show the ground truth contours. The figures show a single example run of each method.

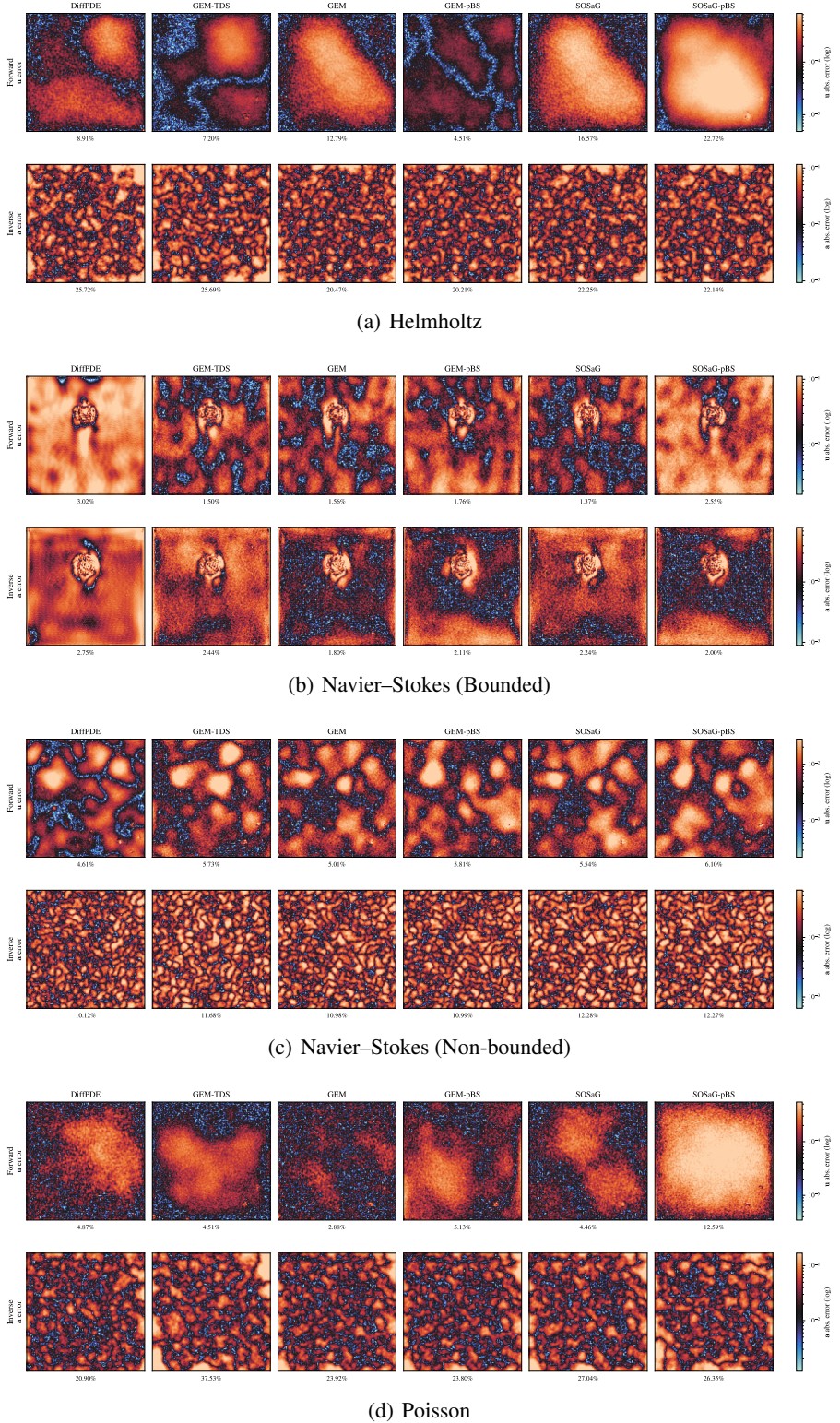

*Figure 8.* Corresponding error comparison panels for all PDE experiments: Helmholtz, Navier–Stokes bounded, Navier–Stokes non-bounded, and Poisson. The errors tend to be more erroneous near the high frequency information regions, this is most obvious in the Bounded Navier–Stokes case.

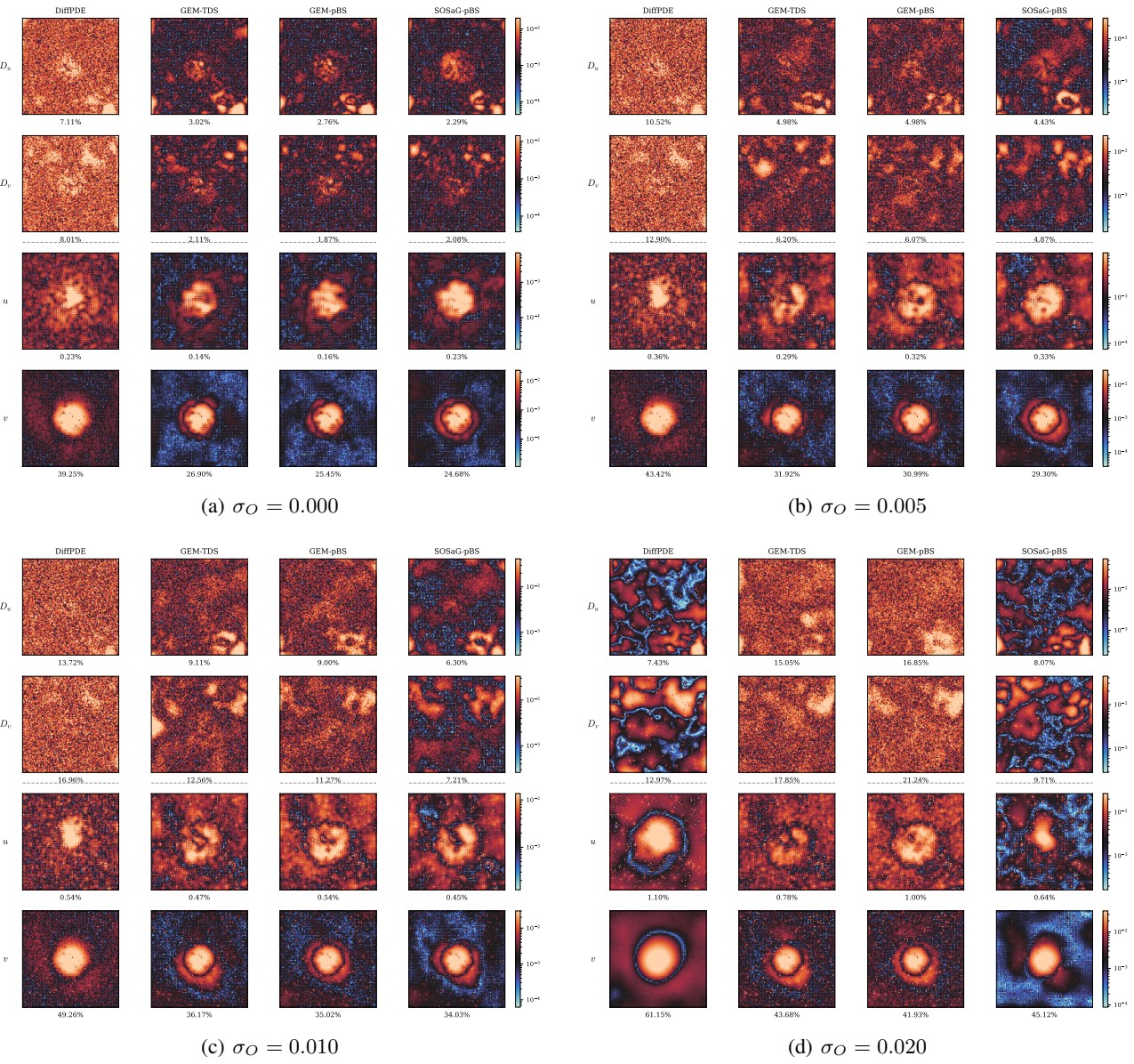

*Figure 9.* Corresponding error comparison panels for the 2SRD PDE system for the reconstructions. The highest error can be observed generally near the high frequency information regions, with this result being more pronounced with increasing levels of observation noise.

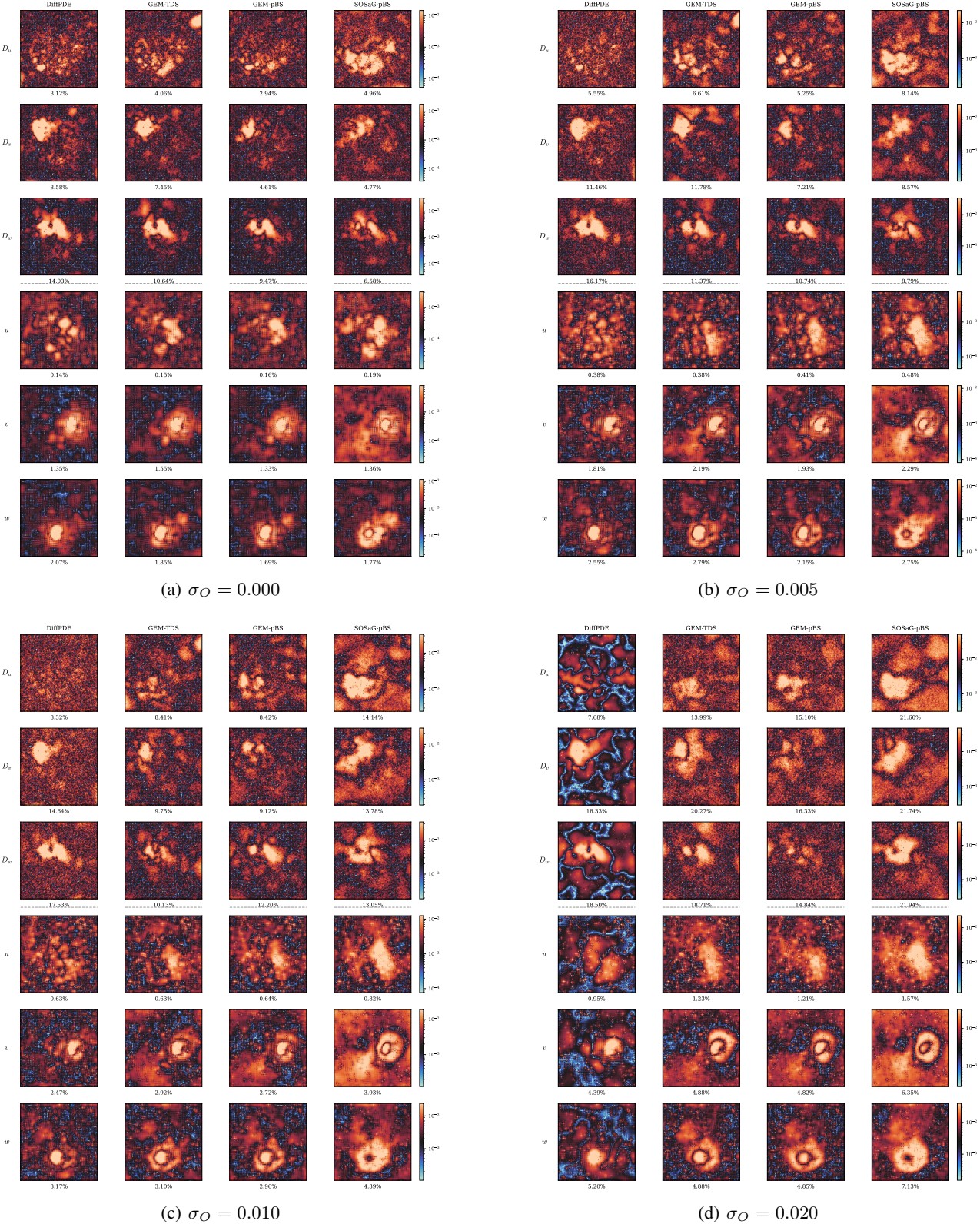

*Figure 10.* Error comparison panels for the 3SRD PDE system. Similarly to Figure 8 and Figure 9, we observe that generally the highest error regions are that near the high frequency information areas with the error increasing as we introduce more observation noise.

