# OpenReview forum: "Particle-Guided Diffusion Models for Partial Differential Equations"
_ICML.cc/2026/Conference — ICML 2026 regular_

### Official Review · Reviewer_Epr8 · 2026-02-17

**Soundness:** 3
**Presentation:** 3
**Significance:** 3
**Originality:** 2
**Overall Recommendation:** 4
**Confidence:** 5

**Summary:**

The paper presents a modified twisted SMC method for PDE simulations. To make use of a second-order proposal, they have to modify the Feynman-Kac mode which introduces some approximation. The method is computationally expensive but appears to provide SOTA results.

**Compliance With Llm Reviewing Policy:**

Affirmed.

**Final Justification:**

Overall, while it is fairly incremental in terms of methodology, it is a good application paper.

**Key Questions For Authors:**

* Could you establish more formal results on the approximate posterior distribution you are targeting? How far is it from the true target?

* Could you come up with a tractable approximation to SOSaG that could be corrected?

* Have you tried cleverer approximation to the likelihood term in equation (11) (which is well know to be rather crude)?

* Have you tried other test-time scaling techniques? It would be very beneficial to add such baselines.

**Limitations:**

* It would be very interesting to discuss the potential failure modes of this method.

**Strengths And Weaknesses:**

Strengths

* While it relies on standard twisted SMC/guidance tools, it's a well-executed application to an interesting problem.

* Contrary to most of the diffusion literature, the paper is clear about the limitations of SMC in this context.

* The experimental results appear to suggest that the method is SOTA on many applications.

Weaknesses

* There is limited methodological innovation compared to previous applications of twisted SMC for diffusions.

* The use of the SOSaG integrator introduces some bias. To address this problem, the authors modify the Feynman-Kac model (typo: pseudo-bootstrap) and the resulting modified target is not the correct posterior distribution. No analysis of the discrepancy between the correct and modified target is provided.

* I don't really buy that SMC behaves as evolutionary algorithms. For N is large, you can make the case (although SMC do not rely on crossover), for N very small... it's really more like some greedy search to me.

* The method is significantly more computationally expensive than existing baselines.

Minor

* Douc & Cappe, 2005 is a strange reference for adaptive resampling, given it does not discuss it. Buchholtz et al. 2021 is another strange reference for the point you are making - standard one is Del Moral et al., 2006. Give also references on early twisted SMC schemes, e.g. Guarneiro et al., 2017.

---

> ### Author Rebuttal · Authors · 2026-03-30
>
> **Response to Question 1**
>
> We thank the reviewer for this question. We have characterised the distance between the tempered and standard target distributions under Gaussian assumptions in our response to Reviewer 2, Question 1, and discuss the broader motivation for likelihood upweighting in our response to Reviewers 1 and 2.
>
> **Response to Question 2**
>
> We thank the reviewer for this suggestion. Constructing a tractable approximation to SOSaG that admits exact importance weight corrections is non-trivial for the following reasons.
>
> One natural option is to assume that the particles produced by the second-order proposal are approximately Gaussian distributed, in which case the TDS framework [1] could be used to compute corrected importance weights. However, we argue this is a crude approximation. The SOSaG proposal first adds noise to the current particle and then passes it through a learned denoiser. The composition of these two operations is unlikely to produce Gaussian-distributed outputs. The denoiser is a highly nonlinear neural network whose output distribution is not analytically tractable.
>
> A second option would be to add a small amount of Gaussian noise to the denoiser output, with the intention of making the particles approximately Gaussian distributed. However, this introduces a problem-dependent hyperparameter that is difficult to set in practice. Furthermore, if the denoiser outputs follow approximately a Student-$t$ distribution, adding Gaussian noise produces a convolution that is not itself Gaussian for any finite noise variance. Making the result approximately Gaussian would require a noise scale large enough to dominate the heavy tails, which would destroy the informative structure the second-order proposal is designed to preserve.
>
> For these reasons, constructing a tractable and correctable approximation to SOSaG remains an open problem, and we identify it as an important direction for future work. We will add this to the limitations section.
>
> **Response to Question 3**
>
> We thank the reviewer for raising this point. More sophisticated likelihood approximations for diffusion-based inverse problems, such as those surveyed in [2], generally require the forward operator to be linear. However, the PDE residual component of our likelihood is nonlinear by construction, it measures how well $x_0$ satisfies a nonlinear PDE, and cannot be expressed as a linear function of $x_0$.
>
> **Response to Question 4**
>
> We thank the reviewer for this suggestion. The most natural test-time scaling baseline in our setting is increasing the number of diffusion steps $K$ for DiffusionPDE. The table below shows that at similar inference time, SOSaG with $N=8, K=2000$ outperforms DiffusionPDE Guided with $K=8000$ across both Darcy flow and Helmholtz, demonstrating that our method achieves better quality-compute tradeoffs than simply scaling DiffusionPDE with more steps.
>
> **Table: Matched inference-time comparison on Darcy flow and Helmholtz.**
>
> | Experiment | Variant | Time (s) | Error Rate $a$ | Relative Error $u$ |
> |---|---|---|---|---|
> | Darcy | DiffPDE Guided $K=2000$ | $276.23 \pm 0.90$ | $0.0347 \pm 0.0022$ | $0.0331 \pm 0.0091$ |
> | Darcy | SOSaG $N=8, K=500$ | $341.07 \pm 0.84$ | $0.0333 \pm 0.0160$ | **0.0112 ± 0.0034** |
> | Darcy | DiffPDE Guided $K=8000$ | $1161.82 \pm 3.14$ | $0.0350 \pm 0.0019$ | $0.0440 \pm 0.0131$ |
> | Darcy | SOSaG $N=8, K=2000$ | $1301.19 \pm 33.85$ | **0.0180 ± 0.0018** | $0.0116 \pm 0.0082$ |
> | Helmholtz | DiffPDE Guided $K=2000$ | $274.38 \pm 0.96$ | $0.1333 \pm 0.0047$ | $0.1661 \pm 0.1023$ |
> | Helmholtz | SOSaG $N=8, K=500$ | $340.26 \pm 0.33$ | $0.1422 \pm 0.0054$ | $0.0303 \pm 0.0018$ |
> | Helmholtz | DiffPDE Guided $K=8000$ | $1109.38 \pm 2.87$ | $0.1375 \pm 0.0053$ | $0.1087 \pm 0.0461$ |
> | Helmholtz | SOSaG $N=8, K=2000$ | $1261.97 \pm 1.28$ | **0.1249 ± 0.0037** | **0.0299 ± 0.0018** |
>
> **References**
>
> [1] Wu, L. et al. (2023). Practical and asymptotically exact conditional sampling in diffusion models. *NeurIPS 2023*.
>
> [2] Daras, Giannis, et al. "A survey on diffusion models for inverse problems."

---

> > ### Author Rebuttal · Reviewer_Epr8 · 2026-04-02
> >
> > The authors have addressed satisfactorily my comments. The methodological contribution remains fairly incremental but this is well-executed.

---

### Official Review · Reviewer_4hYV · 2026-03-05

**Soundness:** 3
**Presentation:** 2
**Significance:** 3
**Originality:** 2
**Overall Recommendation:** 3
**Confidence:** 4

**Summary:**

The paper proposes a guided stochastic sampling framework for solving PDEs using pretrained diffusion models. The method embeds PDE residuals and sparse observations as a likelihood into an SMC framework. Two proposal variants are introduced: Guided Euler-Maruyama and Second-Order Stochastic and Guided. A pseudo-bootstrap weighting scheme is developed to accommodate the non-Gaussian nature of the SOSaG proposal.  Experiments cover five benchmark PDEs and two interacting reaction-diffusion systems.

**Compliance With Llm Reviewing Policy:**

Affirmed.

**Final Justification:**

The added equal-compute and no-resampling results help clarify several of my questions, and I agree the paper shows meaningful empirical improvements. However, after considering the paper and rebuttal as a whole, I still find the methodological contribution somewhat incremental, and I remain unconvinced that the justification for the strongest-performing pBS-based variant is sufficiently principled or well characterized. Therefore, I am maintaining my current score.

**Key Questions For Authors:**

see Strengths And Weaknesses, additionally:
1. Does DiffusionPDE with 4-5x more steps match SMC performance at equal compute?
2. Table~1 shows SOSaG ($N=1$) underperforms DiffPDE on Poisson ($10.23$ vs.\ $8.99$) and non-bounded Navier--Stokes ($7.93$ vs.\ $7.72$), despite being equivalent to DiffusionPDE with stochastic rather than deterministic sampling. This suggests stochastic integration alone is not uniformly beneficial without resampling. This raises a fundamental question: is SOSaG a better proposal because of its second-order construction, or does it merely generate higher-variance particles that give the resampling step more diversity to exploit? If the latter, any high-variance stochastic proposal would achieve similar results and the specific design of SOSaG is incidental.
3. Section3.4 argues the pBS target is preferable because it upweights the PDE likelihood. However, Table1 shows EM(TDS) and EM(pBS) perform nearly identically on four benchmarks, and EM(pBS) is substantially worse on BNS ($4.86$ vs.\ $2.32$). Since both weights are tractable for GEM, this is a clean ablation — the results suggest pBS is a technical workaround for SOSaG's intractable proposal density, not a principled improvement. Can the authors explain the BNS degradation and provide evidence distinguishing these two interpretations?

**Limitations:**

yes

**Strengths And Weaknesses:**

Strengths
1. The SMC wrapper consistently reduces relative error, especially on the solution field u. The gains on Helmholtz and non-bounded NS are meaningful (~3-5x for u).
2. SOSaG handles observation noise better. Under noisy conditions (σO​>0), the second-order correction gives a clear advantage, which is a practically relevant regime.

Weaknesses
1. Novelty is primarily in the combination rather than individual components. The individual building blocks, GEM (Wu et al., 2023), SOSaG (Karras et al., 2022), and pBS (classical bootstrap PF), are all established. The contribution lies in their integration into a physics-informed SMC framework for PDE solving, which is a reasonable and non-trivial engineering effort. However, the paper would benefit from more explicitly articulating what new design insights, beyond empirical validation, this combination produces.
2. pBS justification is circular. The argument in Section 3.4 amounts to "the altered target works better empirically." The choice ρ=1\rho=1
ρ=1 is unjustified. More tellingly, EM (TDS) and EM (pBS) perform nearly identically in Table 1 in four benchmarks, suggesting pBS is a technical workaround for SOSaG's intractable proposal density rather than a principled improvement.
3. No compute-normalized comparison. SMC variants are 4-5x slower (~1200s vs ~280s). It is unclear whether gains come from SMC resampling or simply more function evaluations.

---

> ### Author Rebuttal · Authors · 2026-03-30
>
> **Response to Question 1**
>
> We thank the reviewer for this precise question. The table in our response to Reviewer 4 addresses it directly. At similar inference time, SOSaG with $N=8, K=500$ achieves a relative error of $0.0112 \pm 0.0034$ on Darcy flow in $341$ seconds, compared to $0.0331 \pm 0.0091$ for DiffusionPDE Guided with $K=2000$ in $276$ seconds, a $3\times$ improvement in relative error $u$ at comparable wall-clock time. On Helmholtz, SOSaG similarly outperforms DiffusionPDE at similar compute. We therefore conclude that increasing the number of DiffusionPDE steps does not close the performance gap with SOSaG at equal compute and that SOSaG could conceivably run with shorter inference time with little impact on results.
>
> **Response to Question 2**
>
> We thank the reviewer for this careful observation. We concede that SOSaG with $N=1$ underperforms DiffPDE on Poisson and BNS, and we agree that this is informative. Stochastic integration alone, without resampling, is not uniformly beneficial.
>
> However, we do not advocate for SOSaG as a standalone sampler. Rather, we advocate for using the stochastic second-order proposal as a mutation kernel within an SMC/EA framework, where its benefits are fully realised through the combination of sample diversity and selective resampling. The $N=1$ case isolates the proposal in the absence of the evolutionary mechanism, and mild underperformance in this setting is expected.
>
> This is directly demonstrated by the no-resampling ablation below, which shows a clear hierarchy: DiffPDE < $N$ parallel SOSaG chains without resampling < $N$ chains with resampling. DiffPDE is outperformed by $N$ independent SOSaG chains without resampling, confirming that the second-order proposal generates higher-quality particles than the deterministic baseline. Adding resampling then provides a further significant improvement, confirming that the evolutionary selection mechanism contributes independently on top of proposal quality. This three-way comparison directly distinguishes between the reviewer's two explanations: if it were purely a variance effect, the no-resampling baseline would not outperform DiffPDE, since there is no diversity to exploit without selection. Due to time constraints, we have run this ablation on Darcy flow for now and will extend to other datasets in the final manuscript.
>
> We would also like to note that we do not claim SOSaG is the optimal proposal within our framework, but that our framework allows future work the freedom to choose other integrators best suited to the task at hand.
>
> **Table: SOSaG and EM with and without resampling on Darcy flow.**
>
> | Variant | Time (s) | Error Rate $a$ | Relative Error $u$ |
> |---|---|---|---|
> | DiffPDE Guided $K=8000$ | $1161.82 \pm 3.14$ | $0.0350 \pm 0.0019$ | $0.0440 \pm 0.0131$ |
> | EM $N=8, K=2000$ | $1170.68 \pm 0.56$ | $0.0186 \pm 0.0029$ | $0.0127 \pm 0.0127$ |
> | EM $N=8, K=2000$, No Resampling | $1090.39 \pm 0.85$ | $0.0342 \pm 0.0032$ | $0.0326 \pm 0.0181$ |
> | SOSaG $N=8, K=2000$, No Resampling | $1267.48 \pm 1.18$ | $0.0331 \pm 0.0032$ | $0.0272 \pm 0.0077$ |
> | SOSaG $N=8, K=2000$ | $1301.19 \pm 33.85$ | **0.0180 ± 0.0018** | **0.0116 ± 0.0082** |
>
> **Response to Question 3**
>
> We thank the reviewer for this question. Our original motivation for pBS was to upweight the PDE likelihood, bringing the target closer to the true deterministic solution as discussed in our responses to Reviewers 1 and 2. We acknowledge that EM+pBS does not perform as well as expected. How to choose $\rho$ is dependent on the prior and the likelihood and in this case EM and pBS does not work well together. In this work, we advocate for combining several components into one cohesive framework. The combination of EM and pBS was added as a control to show that pBS must be used in conjunction with a better proposal to produce good results.

---

> > ### Author Rebuttal · Reviewer_4hYV · 2026-04-01
> >
> > Thank you for the detailed rebuttal. The added equal-compute and no-resampling results help clarify several of my questions, and I agree the paper shows meaningful empirical improvements. However, after considering the paper and rebuttal as a whole, I still find the methodological contribution somewhat incremental, and I remain unconvinced that the justification for the strongest-performing pBS-based variant is sufficiently principled or well characterized. Therefore, I am maintaining my current score.

---

### Official Review · Reviewer_VcLr · 2026-03-13

**Soundness:** 2
**Presentation:** 2
**Significance:** 3
**Originality:** 2
**Overall Recommendation:** 4
**Confidence:** 4

**Summary:**

This paper proposes a particle-guided diffusion framework for PDE solving under partial observations. The method starts from a pretrained diffusion prior over joint coefficient/solution fields and adds inference-time guidance from sparse observations and PDE residuals. The paper instantiates this idea in an SMC framework: a guided Euler–Maruyama proposal that matches prior SMC-style conditional diffusion work, and a second-order stochastic guided proposal (SOSaG) that cannot be corrected with the same pointwise density ratio, so the authors introduce a pseudo-bootstrap (pBS) variant that is explicitly biased but claimed to work better empirically. Experiments are reported on several standard PDE benchmarks plus 2-species and 3-species reaction–diffusion systems with noisy observations, with improvements over DiffusionPDE and its unguided variant on the reported relative-error metrics.

**Compliance With Llm Reviewing Policy:**

Affirmed.

**Final Justification:**

While my main concern is still unresolved, I find the presentation and findings of this paper to be important nonetheless. I have raised my score to weak accept.

**Key Questions For Authors:**

1) The strongest-performing method appears to be SOSaG+pBS, but this no longer targets the posterior in Eq. (8); instead it effectively introduces an extra likelihood factor. Can the authors characterize the induced bias more formally, or provide conditions under which this modified target is expected to improve reconstruction quality rather than distort it?

2) The paper’s main direct comparison is to DiffusionPDE, while comparisons to PINNs/FNO/PINO/DeepONet are inherited from the prior DiffusionPDE paper rather than rerun here. Can the authors provide direct same-setup comparisons for at least a small subset of strong baselines on the new reaction–diffusion tasks or under matched observation/noise settings?

3) The paper argues that SMC helps mainly via a multiple-try or evolutionary effect rather than statistical consistency. Can the authors provide targeted evidence for this claim, beyond observing low ESS? For example, comparisons against equal-compute independent guided trajectories without resampling, or analyses relating performance to particle diversity/coalescence, would help.

4) How sensitive are the results to the weights $\beta,\gamma,\omega$, particle count $N$, resampling threshold, and the noise-jitter schedule in SOSaG? Right now the method seems to introduce several design choices, but the robustness story is underdeveloped. If performance is stable, that would improve my confidence in practicality; if not, it would clarify the true scope.

5) Why is it appropriate to describe the method as yielding a scalable generative PDE solver given the substantial increase in runtime reported for the SMC variants? Please clarify whether “scalable” refers to dimensionality, memory, wall-clock, or something else.

**Limitations:**

No. The limitations section discusses compute/runtime and likelihood-cost issues, which is useful, but the impact statement is effectively non-responsive and the paper does not adequately discuss broader limitations or possible failure modes. At minimum, the authors should discuss: dependence on pretrained simulation data; mismatch between the learned prior and the true PDE family; brittleness under observation-model misspecification or out-of-distribution boundary/forcing regimes; and the fact that the best method is biased away from the nominal posterior. The societal-impact discussion can remain brief, but simply saying there is nothing specific to highlight is too weak for a paper positioned as a scientific solver.

**Strengths And Weaknesses:**

The authors examine how to combine diffusion priors, physics-based guidance, and particle methods for PDE inference under sparse observations. Overall, the research analyzes a central concept in scientific machine learning, namely whether one can use pretrained generative priors while still enforcing physical admissibility at test time. This is a relevant problem, and the paper is trying to bridge several active threads: diffusion-based PDE solvers, score-based/data-assimilation-style conditioning, and SMC guidance for diffusion models. DiffusionPDE already established inference-time PDE-residual guidance for partially observed PDE solving, while TDS established SMC correction for conditional diffusion models more broadly; this submission’s main novelty is the attempt to combine those two lines and the use of a stochastic second-order proposal inspired by EDM-style samplers.

The technically clean part of the paper is the GEM/TDS-style construction, which is essentially inherited from prior SMC diffusion conditioning. The more interesting proposed variant, SOSaG+pBS, is not shown to target the desired posterior. In fact, the paper states that the terminal target becomes proportional to an approximate posterior times an extra likelihood factor, and then argues that this may actually be better because the “true target” in practice may differ from the posterior implied by the pretrained prior. That is a heuristic justification, not a principled one. As written, the method’s best-performing variant is not an exact or even clearly controlled approximation to the Bayesian target, and the paper does not quantify this bias or show when it should help rather than hurt. The 'bridging between pBS and PINNs' subsection is more intuition than theory, and the 'evolutionary algorithm' interpretation of degenerate SMC is speculative rather than demonstrated.

A second issue is that the experimental evidence is somewhat narrower than the claims. The paper claims superiority over existing guided generative PDE solvers, but the direct experimental comparison is mostly against DiffusionPDE and an unguided ablation, while stronger non-diffusion baselines are deferred to statements from the DiffusionPDE paper rather than rerun in the present setup. Given that this paper changes the sampler and not the pretrained prior, I understand why DiffusionPDE is the key baseline, but for a convincing case I still wanted at least a limited direct comparison to one or two strong alternatives under the same observation regime, especially on the new reaction diffusion settings. Relatedly, the paper acknowledges a substantial runtime cost, for Darcy-flow ablations, pBS variants are roughly 4–5x slower than DiffPDE-scale inference in the provided timing table. That may be acceptable, but the significance case should then be made more explicitly in terms of accuracy-per-time or robustness-per-time.

Some notation is sloppy in a way that makes technical auditing harder than it should be. For example, the likelihood in Eq. (10) is written using masks applied directly to $x$, even though $x$ concatenates coefficient and solution fields and the notation should distinguish the relevant components more carefully. There are multiple writing issues and some claims are stronger than the supporting evidence.

I view the paper as moderately incremental rather than highly novel. DiffusionPDE already handled PDE-residual guidance for generative PDE solving under partial observation. TDS and related work already introduced SMC-based conditioning for diffusion models. Score-based data assimilation and conditional diffusion for PDE simulation are also established. The paper’s real new ingredient is combining these ideas with a stochastic second-order proposal and then relaxing exactness through the pBS construction. That is a legitimate contribution, but not a fundamentally new problem formulation or paradigm.

---

> ### Author Rebuttal · Authors · 2026-03-30
>
> **Response to Question 1**
>
> We thank the reviewer for this question. Full derivations will be included in the revised manuscript; we summarise the key results here.
>
> It is first worth emphasising that the goal of this framework is not uncertainty quantification but accurate numerical approximation of a deterministic PDE solution. The true solution generating distribution is therefore degenerate. Since a PDE solution is deterministic, the ideal physical target is a Dirac measure concentrated on the true solution, corresponding to an exact likelihood with an uninformative prior. The standard Bayesian formulation $p_\theta(x_0) \cdot \tilde{p}_\theta(y \mid x_0)$ used in works such as DiffPDE is not the true physical model but an ansatz that introduces the prior as a computational convenience, softening the hard constraint of the true Dirac delta target. Our tempered target likelihood$^{\rho}$ $\times$ prior is a further design choice along this spectrum, by construction it concentrates mass more tightly around solutions satisfying the PDE constraints, sitting closer to the true deterministic target than the untempered baseline. The prior uncertainty in this setting is an artefact of the probabilistic framing rather than a reflection of genuine solution uncertainty, which motivates upweighting the likelihood to partially correct for this.
>
> The SOSaG+pBS method targets the tempered posterior with $\rho = 2$. In the Gaussian case the bias is analytically tractable: SOSaG+pBS has strictly smaller posterior covariance than the standard posterior, corresponding to an overestimation of measurement precision equivalent to halving the likelihood noise variance. The mean is shifted towards the observations. Formally, the precision bias is $\Sigma_{\text{SOS}}^{-1} - \Sigma_{\text{EM}}^{-1} = \sigma_\ell^{-2} A^\top A \geq 0$.
>
> In the general case where $p_\theta$ is a diffusion model prior and the posterior is potentially multimodal, closed-form expressions for this are not available. This is compounded by the non-linear residual term in our likelihood. However, the qualitative effect of upweighting the likelihood remains: the tempered posterior concentrates mass more tightly around regions of high likelihood, which is desirable in the PDE setting where the true solution is deterministic.
>
> The tempering ablation is provided in our response to Reviewer 1. Moderate tempering yields consistent improvements, while degradation at low values of $\rho$ is consistent with the broader Bayesian literature [1, 2], and the parameter requires tuning per experiment. Likelihood tempering has also been independently investigated in [3], Appendix D.2.1, as the *twist scale*.
>
> **Response to Question 2**
>
> We thank the reviewer for this suggestion. Within the review period we have prioritised the experiments that we believe provide the most informative evidence. During the discussion period we will endeavour to obtain further comparisons results, having prioritised new experiments responding to all Reviewers within our limited time.
>
> **Response to Question 3**
>
> The ablation in Table 5 in the manuscript shows that higher particle count $N$ and tempering $\rho > 1$ consistently improve reconstruction quality. The $\rho > 1$ regime induces collapsing ESS, reducing the particle system to a structured multiple-try search. The connection between Feynman-Kac models and EAs has been established in prior work [4], and we will formalise the specific connection to the $(1+\lambda)$ EA in the revised manuscript.
>
> Following the reviewer's suggestion, we ran a Best-of-$N$ baseline without resampling. Full results are provided in our response to Reviewer 3. Resampling meaningfully outperforms this baseline, confirming that intermediate selection contributes beyond the pure multiple-try effect and that the SMC structure is doing useful work.
>
> **Response to Questions 4 & 5**
>
> We agree that "scalable" was imprecise and will revise this language in the manuscript. The noise schedule follows [5]; the resampling threshold is a null choice in the weight-degenerate regime since we resample at every iteration; and the particle count ablation is provided in Appendix Table 5.
>
> On the question of runtime, matched inference-time comparisons showing that SOSaG achieves superior reconstruction quality to DiffusionPDE under comparable compute budgets are provided in our response to Reviewer 4.
> **References**
>
> [1] Aitchison, L. (2021). A statistical theory of cold posteriors in deep neural networks. *ICLR 2021*.
>
> [2] Wenzel, F. et al. (2020). How good is the Bayes posterior in deep neural networks really? *ICML 2020*.
>
> [3] Wu, L. et al. (2023). Practical and asymptotically exact conditional sampling in diffusion models. *NeurIPS 2023*.
>
> [4] Moral, P. (2004). Feynman-Kac formulae: genealogical and interacting particle systems with applications. Springer New York.
>
> [5] Huang et al. (2024). DiffusionPDE: Generative PDE-solving under partial observation. *NeurIPS 2024*.

---

> > ### Author Rebuttal · Reviewer_VcLr · 2026-04-03
> >
> > Thank you for the thoughtful rebuttal. It addresses part of my concern, but not fully.
> >
> > The response to Q1 improves the framing. I now better understand the authors’ position that the goal is accurate recovery of a deterministic PDE solution, not calibrated uncertainty quantification, and that the Bayesian posterior is being used as a computational surrogate rather than a literal physical target. The Gaussian example is also helpful in making the effect of tempering more concrete.
> >
> > My main concern remains only partially resolved because the justification for the tempered target is still primarily heuristic. The rebuttal argues that a more likelihood-concentrated target may be preferable in deterministic PDE settings. However, this is still a modeling choice rather than a principled derivation from the original posterior. I would therefore like the revised paper to state this point very clearly and avoid language that suggests posterior exactness for the strongest variant.
> >
> > The response to Q3 is useful. The added Best-of-N comparison is important, and if resampling outperforms Best-of-N under comparable compute, that strengthens the claim that the SMC structure contributes beyond simple multiple-try search. This evidence should be presented clearly in the revision, since it is central to the paper’s conceptual case.
> >
> > The responses to Q2 and Q4/Q5 are reasonable, but these concerns remain only partially addressed. In particular, the empirical case would be stronger with direct same-setup comparisons beyond DiffusionPDE where feasible, and the significance claim depends on the matched-compute results being clearly shown.
> >
> > Overall, the rebuttal improves the framing and helps clarify the intent of the method, but my main concerns are still only partially resolved.

---

### Official Review · Reviewer_Jkww · 2026-03-13

**Soundness:** 2
**Presentation:** 3
**Significance:** 2
**Originality:** 2
**Overall Recommendation:** 3
**Confidence:** 2

**Summary:**

The manuscript proposes to use the SMC guidance in diffusion models to solve PDE inverse problems. A new SMC proposal distribution is proposed to utilize a more accurate SDE integrator, named SOSaG. There are theoretical inconsistencies in the proposed SOSaG, which are further resolved via approximated proposal density. The proposed method is evaluated on PDE inverse problem benchmarks and shows improved performance over conventional diffusion-based PDE inverse problem solvers and vanilla SMC-guided diffusion-based PDE solvers.

**Compliance With Llm Reviewing Policy:**

Affirmed.

**Final Justification:**

I think this submission needs a clearer framing. Currently, it seems ambiguous whether the problem is a PDE-constrained optimization or a Bayesian inverse problem. Thus, the motivation and the design of the proposed algorithm should be better justified to address my concern. I maintain my original assessment.

**Key Questions For Authors:**

See weaknesses.

**Limitations:**

Yes

**Strengths And Weaknesses:**

Strengths

1. The proposed method utilizes SMC guidance, which is theoretically more justified than conventional diffusion guidance. The application to PDE inverse problems is a contribution to the sciML community.

Weaknesses

1. The authors motivate the usage of SOSaG by its increased accuracy compared to the E-M scheme (Sec. 3.3), which targets a more accurate prior distribution (the diffusion). However, in Sec 3.4, the authors claim that emphasizing more on the likelihood (PDE residual & observation) can be beneficial as the prior is unreliable, thereby justifying approximating particle weights. These two statements seem contradictory to me.

---

> ### Author Rebuttal · Authors · 2026-03-30
>
> We thank the reviewer for this observation, which we agree deserves clarification. The two claims operate at different levels of the algorithm. The choice of integrator concerns *numerical accuracy* in simulating a chosen target, while the choice of target concerns *statistical modelling* of the prior. These are independent design decisions.
>
> This framework is not primarily intended for uncertainty quantification; its goal is to obtain a numerical approximation to a deterministic PDE solution using a Bayesian framework. The ideal target would be an exact likelihood (the perfect PDE operator with infinitesimal discretisation) with an uninformative prior, corresponding to solving the PDE exactly. However, this is intractable in practice, and sampling directly from such a target is computationally infeasible.
>
> To make the problem tractable, we define a target distribution that is computationally amenable whilst remaining close to the true physical target. Following DiffusionPDE, we introduce a diffusion prior, giving a target of the form likelihood $\times$ prior. However, in this context the true physical PDE solution would correspond to a likelihood given by a Dirac measure (a hard constraint). The likelihood model should be seen as a design choice that balances computational tractability for softness of the physics constraints. From this point of view, tempering the likelihood and targeting likelihood$^{\rho}$ $\times$ prior, as is the approximate effect of the pBS framework, simply results in a different design choice.
>
> The SOSaG integrator is target agnostic. Regardless of which target we sample from, a more accurate numerical integrator will better approximate it. This is supported empirically in [2, 3], where second-order schemes consistently outperform first-order methods across a range of tasks.
>
> The argument in Section 3.4 concerns the *target itself* rather than the integrator. Since $p_\theta$ is an imperfect approximation of the true data distribution, the target should be adjusted by upweighting the likelihood. With Gaussian priors, a tempered posterior with $\rho > 1$ is exactly equivalent to a cold posterior with a rescaled prior variance, directly correcting for prior overdispersion. This is consistent with [4, 1], which shows that under prior misspecification, likelihood temperatures $\rho > 1$ can reduce errors. Because the likelihood is itself approximate, one cannot simply take $\rho \to \infty$, as the approximation error in the likelihood would then dominate, plus it would be difficult to sample from a point mass distribution. The parameter $\rho$ therefore requires tuning per experiment, and we will discuss annealing $\rho$ in the limitations section. When the prior is unknown and the likelihood involves non-linear operations, it is impossible to exactly characterise the difference between the upweighted and original targets, we therefore provide a characterisation under Gaussian assumptions in response to Reviewer 2.
>
> Our ablation over $\rho$ in Table 5 of our manuscript confirms that $\rho > 1$ improves reconstruction quality across both sampling methods. We have expanded this ablation below.
>
> **Table: Tempering ablation on Darcy flow averaged over 20 runs.$N=8, K=500$**
>
> | Variant | Error Rate $a$ | Relative Error $u$ |
> |---|---|---|
> | SOSaG $\rho=1000$ | $0.0246 \pm 0.0035$ | $0.0075 \pm 0.0010$ |
> | SOSaG $\rho=100$ | $0.0247 \pm 0.0037$ | $0.0074 \pm 0.0008$ |
> | SOSaG $\rho=10$ | $0.0210 \pm 0.0024$ | $0.0066 \pm 0.0008$ |
> | SOSaG $\rho=1$ | $0.0199 \pm 0.0024$ | $0.0112 \pm 0.0045$ |
> | SOSaG $\rho=0.1$ | $0.0432 \pm 0.0060$ | $0.3596 \pm 0.0687$ |
> | SOSaG $\rho=0.01$ | $0.1641 \pm 0.0359$ | $1.8516 \pm 0.2157$ |
> | SOSaG $\rho=0.001$ | $0.1495 \pm 0.0475$ | $2.7168 \pm 0.2068$ |
>
> These results will be added to the revision. It is also worth noting that [5], Appendix D.2.1 investigated tempering under the name *twist scale*, finding higher values useful for class-conditional sampling.
>
> In summary, the accuracy of the integrator and the choice of target are separable concerns, each motivated by independent arguments. We will add a clarifying sentence to Section 3.3 to make explicit that the integrator choice is target agnostic, and revise Section 3.4 to emphasise that the prior reliability argument concerns the choice of target rather than the integrator.
>
> **References**
>
> [1] Aitchison, L. (2021). A statistical theory of cold posteriors in deep neural networks. *ICLR 2021*.
>
> [2] Karras, T. et al. (2022). Elucidating the design space of diffusion-based generative models. *NeurIPS 2022*.
>
> [3] Pierret, E., & Galerne, B. (2025). Diffusion models for Gaussian distributions: Exact solutions and Wasserstein error. *ICML 2025*.
>
> [4] Wenzel, F. et al. (2020). How good is the Bayes posterior in deep neural networks really? *ICML 2020*.
>
> [5] Wu, L. et al. (2024). Practical and asymptotically exact conditional sampling in diffusion models. *NeurIPS 2023*.

---

> > ### Author Rebuttal · Reviewer_Jkww · 2026-04-01
> >
> > I acknowledge the authors for the detailed reply, and the concern about the technical details has been addressed.
> > However, I have follow-up questions/comments regarding the problem setting of the paper. (Bayesian sampling vs PDE solving)
> >
> > 1. If the goal is to solve PDEs, the exact problem setup should be more explicitly stated in the beginning and also in Eq. 8.
> > 2. It seems that comparison to deterministic PDE solvers using the PDE residual information solely would then become necessary to demonstrate the effectiveness of the proposed method. E.g.,  standard PINNs (showing the diffusion prior is indeed effective, rather than harming the PDE residual-based guidance) or other neural operators (demonstrate the advantage of using diffusion models). These methods should also be suitable for deterministic data assimilation (DA), where reconstruction of the full posterior distribution is not required.
> > 3. I believe the core value of applying diffusion models to PDE inverse problems lies in the high-dimensional probabilistic modeling capability. Using energy-guided diffusion (for example, SMC guidance), which is fundamentally posterior sampling, seems to contradict the deterministic PDE solving or data assimilation setting. Are maximum a posteriori-oriented guided diffusion methods more suitable for the problem (e.g., [1,2])?
> >
> > [1] Ben-Hamu et al., D-Flow: Differentiating through Flows for Controlled Generation, ICML'24
> > [2] Zhang et al., Flow Priors for Linear Inverse Problems via Iterative Corrupted Trajectory Matching, NIPS'24
> >
> >
> > minor: line 82 "The guidance methods uses in these framework are not..." -> "The guidance methods used in these frameworks are not..."

---

### Decision · Program_Chairs · 2026-04-30

**Decision:**

Accept (regular)

**Comment:**

The paper studies diffusion-based PDE solving under partial observations by combining physics-based guidance with particle-based sampling. The authors examine an important question: whether a pretrained generative prior can be used with PDE residuals and sparse observations to improve reconstruction quality. Across the discussion, the work is generally seen as relevant to scientific machine learning and as a meaningful attempt to connect diffusion-based PDE solvers, physics-informed guidance, and Sequential Monte Carlo methods. The empirical results are promising, with several reviewers noting better reconstruction quality than the main diffusion-based baseline, especially for solution fields and in noisy settings. The paper also appears to be technically sound overall, and the proposed framework is likely to be of interest to researchers working on generative methods for PDE inference.

The main reservations concern the framing and justification of the strongest variant. In particular, the pseudo bootstrap construction appears to trade posterior exactness for empirical performance, and the current explanation remains heuristic. Reviewers also noted that the paper should more clearly distinguish between Bayesian posterior sampling and deterministic PDE solving, and that the empirical case would be stronger with broader baselines using exactly the same setup and clearer, computationally normalized comparisons. Even so, the concerns seem more about clarity and strengthening the justification than about a lack of value in the core contribution. Overall, the research proposed an important idea and presents a solid application paper with useful empirical findings, though the final version would benefit from a more precise statement of its objective and limitations.